# Structural basis for centromere maintenance by *Drosophila* CENP-A chaperone CAL1

Bethan Medina-Pritchard[1] (iD), Vasiliki Lazou[1], Juan Zou[1], Olwyn Byron[2], Maria A Abad[1], Juri Rappsilber[1,3] (iD), Patrick Heun[1] & A Arockia Jeyaprakash[1,*,†] (iD)

## Abstract

Centromeres are microtubule attachment sites on chromosomes defined by the enrichment of histone variant CENP-A-containing nucleosomes. To preserve centromere identity, CENP-A must be escorted to centromeres by a CENP-A-specific chaperone for deposition. Despite this essential requirement, many eukaryotes differ in the composition of players involved in centromere maintenance, highlighting the plasticity of this process. In humans, CENP-A recognition and centromere targeting are achieved by HJURP and the Mis18 complex, respectively. Using X-ray crystallography, we here show how *Drosophila* CAL1, an evolutionarily distinct CENP-A histone chaperone, binds both CENP-A and the centromere receptor CENP-C without the requirement for the Mis18 complex. While an N-terminal CAL1 fragment wraps around CENP-A/H4 through multiple physical contacts, a C-terminal CAL1 fragment directly binds a CENP-C cupin domain dimer. Although divergent at the primary structure level, CAL1 thus binds CENP-A/H4 using evolutionarily conserved and adaptive structural principles. The CAL1 binding site on CENP-C is strategically positioned near the cupin dimerisation interface, restricting binding to just one CAL1 molecule per CENP-C dimer. Overall, by demonstrating how CAL1 binds CENP-A/H4 and CENP-C, we provide key insights into the minimalistic principles underlying centromere maintenance.

**Keywords** CAL1; CENP-A; CENP-C; centromeres; chromosome segregation
**Subject Categories** Cell Cycle; Chromatin, Transcription & Genomics; Structural Biology
**The EMBO Journal (2020) 39: e103234**

## Introduction

Centromeres are specialised chromosomal regions that act as a platform for the assembly of kinetochores, the microtubule anchoring sites essential for chromosome segregation during mitosis and meiosis (Musacchio & Desai, 2017). Unlike budding yeast where DNA sequence is sufficient to define centromere identity, centromeres in most other eukaryotes are defined by the enrichment of unique nucleosomes containing the histone H3 variant CENP-A (Sekulic & Black, 2012; Zasadzinska & Foltz, 2017). As a consequence, maintenance of CENP-A-containing nucleosomes is essential for preserving centromere identity through generations of cell cycles. This is achieved through an epigenetic mechanism that relies on CENP-A as an epigenetic mark (Westhorpe & Straight, 2014; McKinley & Cheeseman, 2016; Musacchio & Desai, 2017; Zasadzinska & Foltz, 2017).

Unlike canonical chromatin maintenance, centromeric chromatin maintenance is decoupled from DNA replication. As a result, CENP-A levels on the sister chromatids are reduced by half during replication (Jansen *et al*, 2007; Hemmerich *et al*, 2008; Dunleavy *et al*, 2009; Mellone *et al*, 2011; Lidsky *et al*, 2013). To ensure stable centromere maintenance, CENP-A nucleosomes must return to their original levels through active CENP-A deposition. The timing of CENP-A deposition varies among species; however, the underlying mechanisms appear to share significant similarity (Zasadzinska & Foltz, 2017). A central player in this process is the CENP-A-specific chaperone HJURP in human and its homologue Scm3 in fungi (Kato *et al*, 2007; Foltz *et al*, 2009; Pidoux *et al*, 2009; Sanchez-Pulido *et al*, 2009; Dunleavy *et al*, 2011). Both HJURP and Scm3 can bind the CENP-A–histone H4 (CENP-A/H4) heterodimer in its pre-nucleosomal form, and these complexes are then targeted to centromeres by the Mis18 complex (Fujita *et al*, 2007; Moree *et al*, 2011; Dambacher *et al*, 2012; Hayashi *et al*, 2014; McKinley & Cheeseman, 2014; Nardi *et al*, 2016; Stellfox *et al*, 2016; French *et al*, 2017; Hori *et al*, 2017). While the human Mis18 complex is composed of Mis18α, Mis18β and Mis18BP1, the fission yeast Mis18 complex consists of Mis18, Mis16, Eic1 and Eic2, where Eic1 and Eic2 are proposed to be functional equivalents of human Mis18BP1 (Fujita *et al*, 2007; Hayashi *et al*, 2014; Subramanian *et al*, 2014). The timing of Mis18 complex assembly, its centromere targeting, and subsequent CENP-A deposition are suggested to be tightly controlled by the kinase activities of CDK and Plk1 (Silva *et al*, 2012; McKinley & Cheeseman, 2014; Stankovic *et al*, 2017; French & Straight, 2019). While we know the identity of key players involved in centromere maintenance, molecular and mechanistic understanding of their

---

1 Wellcome Centre for Cell Biology, University of Edinburgh, Edinburgh, UK
2 School of Life Sciences, University of Glasgow, Glasgow, UK
3 Institute of Biotechnology, Technische Universität Berlin, Berlin, Germany
*Corresponding author. Tel: +44 1316 507113; E-mail: jeyaprakash.arulanandam@ed.ac.uk
†Lead author

intermolecular cooperation are just emerging (Nardi *et al*, 2016; Stellfox *et al*, 2016; Pan *et al*, 2017; Spiller *et al*, 2017).

Strikingly, *Drosophila* species have regional centromeres defined by the presence of CENP-A (also called CID in this organism), but lack clear homologues of HJURP and the subunits of the Mis18 complex. Instead, fly-specific CAL1 appears to combine the roles of both HJURP and the Mis18 complex: pre-nucleosomal CENP-A recognition and its targeting to the centromere for deposition, respectively (Phansalkar *et al*, 2012). Targeting CAL1 to non-centromeric DNA in *Drosophila* cells can recruit CENP-A and establish centromeres capable of assembling kinetochore proteins and microtubule attachments (Chen *et al*, 2014). These observations and the ability of CAL1 to bind CENP-A/H4 and CENP-C with its N- and C-terminal regions, respectively, collectively established CAL1 as a "self-sufficient" CENP-A-specific assembly factor in *Drosophila* (Schittenhelm *et al*, 2010; Chen *et al*, 2014). However, structure-level mechanistic understanding of how CAL1 binds CENP-A/H4 and CENP-C to facilitate the establishment and maintenance of centromeres is yet to be determined. The simplistic nature of the centromere maintenance pathway in *Drosophila* makes it a unique model system to understand the fundamentally conserved structural principles underlying centromere maintenance.

In this study, we present the structural basis for the recognition of CENP-A/H4 and CENP-C by CAL1. Our analysis reveals that although CAL1 does not share noticeable sequence similarity with its human or fission yeast counterpart, it recognises CENP-A/H4 using both conserved and adaptive structural principles. We also provide the structural framework of interactions responsible for CENP-C recognition by CAL1. Our structural analysis, together with validation of structure-guided mutants *in vitro* and in cells, provides the molecular basis for the mechanism by which CAL1 single-handedly recognises and targets CENP-A to centromeres to maintain centromere identity in flies.

# Results

## The N-terminal region of CAL1 forms a heterotrimer with the histone fold domain of CENP-A and H4

Secondary structure prediction analysis indicated that CAL1 is likely to be a predominantly unstructured protein, although it includes an N-terminal domain spanning amino acid (aa) residues 1–200 predicted to fold into α helices (Fig EV1A and B). With the aim of structurally characterising the intermolecular interactions responsible for CAL1 binding to CENP-A/H4, we reconstituted a protein complex containing the N-terminal 160 aa of CAL1, a putative histone fold domain of CENP-A and H4 (His-CAL1$_{1-160}$–CENP-A$_{101-225}$–H4) (Fig 1A) using recombinant proteins as previously reported (Chen *et al*, 2014). Limited proteolysis experiments performed on CAL1$_{1-160}$–CENP-A$_{101-225}$–H4 complex using different proteases suggested that a CENP-A fragment containing aa 144–255 (CENP-A$_{144-255}$) is sufficient to interact with CAL1 and H4. Subsequently, using CAL1$_{1-160}$, CENP-A$_{144-255}$ and H4, we reconstituted a truncated protein complex (His-CAL1$_{1-160}$–CENP-A$_{144-225}$–H4). The molecular weights (MW) measured for His-CAL1$_{1-160}$–CENP-A$_{101-225}$–H4 and His-CAL1$_{1-160}$–CENP-A$_{144-225}$–H4 using size-exclusion chromatography combined multi-angle light scattering (SEC-MALS) are

$47.0 \pm 0.9$ and $43.4 \pm 0.8$ kDa, respectively (Fig EV1C). These values match with calculated MW for a 1:1:1 heterotrimeric assembly for both complexes (46.7 and 41.7 kDa, respectively) and are in agreement with our previous report (Roure *et al*, 2019). This observation is also in agreement with the subunit stoichiometry of the human pre-nucleosomal CENP-A/H4 in complex with HJURP (Hu *et al*, 2011).

## Structure determination of the CAL1$_{1-160}$–CENP-A/H4 complex

Extensive crystallisation trials with CAL1$_{1-160}$–CENP-A$_{101-225}$–H4 and CAL1$_{1-160}$–CENP-A$_{144-225}$–H4 yielded two different crystal forms: form I that diffracted X-rays to about 3.5 Å and form II that diffracted anisotropically to about 4.4 Å (Table 1). Molecular replacement was performed for the dataset collected from form I using the coordinates of *Drosophila melanogaster* (*dm*) H3/H4 heterodimer (deduced from the structure of *dm* nucleosome core particle, PDB: 2PYO) (Clapier *et al*, 2008). Molecular replacement solution yielded initial phases sufficient for subsequent rounds of model building and refinement (Fig EV2A). The final model included residues 17–47 of CAL1, 147–220 of CENP-A and 27–98 of H4 and was refined to an R factor 27.2% and $R_{free}$ factor 28.6% (Fig 1B and Table 1). Although we used a CAL1 fragment spanning residues 1–160 in the crystallisation experiment, the calculated electron density map accounted only for CAL1 residues 17–47. Considering these crystals took more than a year to form, we concluded that CAL1 was proteolytically cleaved, which may have facilitated the crystallisation of a truncated complex.

The refined model obtained using crystal form I was used as a template in molecular replacement to determine the structure of crystal form II (Figs 1C and EV2B). The difference electron density map calculated using the molecular replacement solution revealed unambiguous density for most main chain atoms of CAL1$_{1-160}$. Considering the modest resolution of the structure, intermolecular interactions stabilising the CAL1–CENP-A/H4 complex were further analysed using chemical cross-linking mass spectrometry (CLMS). Purified recombinant CAL1$_{1-160}$–CENP-A$_{101-225}$–H4 complex was cross-linked using EDC (solid lines), a zero-length cross-linker that covalently links carboxylate groups of Asp or Glu residues with primary amines of Lys and N-terminus, or hydroxyl group of Ser, Thr and Tyr, or BS$^3$ (dashed lines), a cross-linker that covalently links amine to amine or hydroxyl group of Ser, Thr and Tyr. The cross-linked peptides were analysed by mass spectrometry to identify intra- and intermolecular contacts (Fig EV3). Notably, the data revealed intramolecular cross-links between the N- and C-terminal regions of CAL1$_{1-160}$, particularly between Ser19 and Lys20 and Glu139 and Glu155, suggesting a direct interaction between these regions (Fig EV3). This information was particularly helpful in tracing the backbone atoms of residues beyond CAL1 residue 47 within the electron density map.

## Overall structure of CENP-A/H4 assembly

The structures obtained from two different crystal forms together provide key insights into the overall architecture of the assembly (Fig 1B and C). Structural superposition analysis showed that CENP-A/H4 heterodimer (form I) aligns well with H3/H4 heterodimer (PDB: 2PYO) with a root mean square deviation (RMSD) of

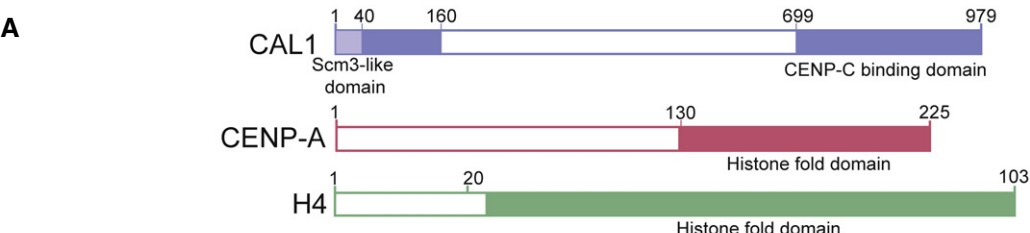

**A** Schematic representation of structural features of CAL1, CENP-A and H4. Filled boxes represent folded domains.

**Figure 1. N-terminal 160 amino acids of CAL1 wrap around CENP-A/H4 heterodimer to form a heterotrimeric assembly.**

A Schematic representation of structural features of CAL1, CENP-A and H4. Filled boxes represent folded domains.
B Overall structure of His-CAL1$_{1–160}$–CENP-A$_{101–225}$–H4 (crystal form I). CAL1 is shown in blue, CENP-A in maroon and H4 in green.
C Overall structure of His-CAL1$_{1–160}$–CENP-A$_{144–225}$–H4 (crystal form II). CAL1 is shown in blue, CENP-A in maroon and H4 in green.

1 Å (Fig EV4A). This suggests that both H3 and CENP-A use an identical mode of H4 binding. However, CENP-A α1, H4 α3 and C-terminal tail show conformational variations in the CAL1-bound CENP-A/H4 complex, likely due to CAL1 binding (Fig EV4B). Particularly, in the H3/H4 structure, the C-terminal tail of H4 folds back and makes contacts with the H3 α3, resembling CAL1 interaction at

**Table 1.  Data collection and refinement statistics.**

| | CAL1$_{1-160}$–CENP-A/H4 Form I | CAL1$_{1-160}$–CENP-A/H4 Form II | CENP-C$_{1264-1411}$ | CAL1$_{841-979}$–CENP-C$_{1264-1411}$ |
|---|---|---|---|---|
| **Data collection** | | | | |
| Space group | R 3 2 :H | P 63 2 2 | P 41 21 2 | P 21 21 21 |
| **Cell dimensions** | | | | |
| $a, b, c$ (Å) | 178.25, 178.25, 133.28 | 199.70, 199.70, 76.75 | 57.20, 57.20, 92.96 | 86.27, 86.44, 88.46 |
| $\alpha, \beta, \gamma$ (°) | 90, 90, 120 | 90, 90, 120 | 90, 90, 90 | 90, 90, 90 |
| Wavelength | 0.97625 | 0.91587 | 0.97623 | 0.97625 |
| Resolution (Å) | 66.79–3.47 (3.59–3.47) | 172.9–4.38 (5.19–4.38) | 28.6–1.82 (1.88–1.82) | 28.9–2.27 (2.35–2.27) |
| $R_{merge}$ | 0.147 (1.96) | 0.129 (0.95) | 0.0548 (0.598) | 0.076 (0.663) |
| $R_{pim}$ | 0.058 (0.759) | 0.046 (0.526) | 0.013 (0.177) | 0.022 (0.202) |
| $I/\sigma I$ | 10.46 (1.15) | 9.61 (2.4) | 34.13 (3.55) | 21.16 (3.40) |
| Completeness (%) | 99.93 (100.00) | 89 (81)[a] | 99.73 (97.87) | 98.41 (94.61) |
| Redundancy | 7.6(7.7) | 8.9 (8.8) | 18.0 (11.9) | 13.1 (11.3) |
| **Refinement** | | | | |
| No. of reflections | 80,934 (8,111) | 20,689 (307) | 14,426 (1,381) | 404,275 (33,273) |
| $R_{work}$ (%)/$R_{free}$ (%) | 27.2/28.6 | 30.6/32.3 | 19.4/23.5 | 23.7/26.6 |
| **No. of atoms** | | | | |
| Protein | 2,803 | 1,715 | 1,065 | 4,229 |
| **Average B** | | | | |
| Protein | 136.6 | 193 | 40.8 | 71.1 |
| **R.m.s deviations** | | | | |
| Bond length (Å) | 0.004 | 0.005 | 0.006 | 0.006 |
| Bond angles (°) | 0.87 | 1.2 | 085 | 1.20 |
| **Ramachandran values** | | | | |
| Favoured (%) | 90.3 | 89.7 | 97.79 | 97.1 |
| Disallowed (%) | 1.2 | 1.7 | 0.00 | 0.19 |

Statistics for the highest-resolution shell are shown in parentheses.
[a]Ellipsoidal completeness (%) from STARANISO (see also Materials and Methods).

the equivalent region of CENP-A in the CAL1/CENP-A/H4 structure. The H4 C-terminal tail possibly swings away from this site upon CAL1 binding. Overall structure of *dm* CENP-A/H4 (form I) is very similar to human CENP-A/H4 (PDB: 3NQJ) (Sekulic *et al*, 2010) with a RMSD of 1 Å (Fig EV4C). However, noticeable conformational variation is seen in loop L1, possibly to accommodate the amino acid variations between HJURP and CAL1 (Fig EV4C).

**CAL1 binds CENP-A/H4 heterodimers through multiple physical contacts**

CAL1$_{1-160}$ is almost entirely made of α helices that make multiple contacts with CENP-A/H4 heterodimer by wrapping around it (Figs 1C and 2A). Most CENP-A contacts are made by CAL1 helices α1 and α2 and loop L1, which interact with the CENP-A helices α2, α1 and loop L1, respectively, involving a total interface area of about 940 Å². Particularly, while the N-terminal half of the CAL1 α1 helix packs against CENP-A α2 involving electrostatic (CAL1 R18 with CENP-A Q90) and hydrophobic (involving CAL1 L11 and M14) interactions, the C-terminal half, mainly aa W22 and F29, is sandwiched between CENP-A α2 and H4 α3 (Fig 2A). CAL1 L1 crosses

over CENP-A L1 to facilitate CAL1 α2 interaction with CENP-A α3. In addition, CAL1 α4 contacts both CENP-A α2 and α3 involving an interface area of about 80 Å². These CAL1–CENP-A interactions appear to be further stabilised by CAL1 α5 and α6 which together with CAL1 α1 make an intramolecular helical bundle resembling a latch that restrains the position of α1 helix (Fig 1C).

**Hydrophobic interactions involving CAL1 W22 and F29 are critical for CENP-A/H4 binding**

Considering the extent of contacts made by the N-terminal 50 aa of CAL1, we checked whether CAL1$_{1-50}$ is sufficient to interact with CENP-A/H4. Using recombinant His-CAL1$_{1-50}$, H4 and CENP-A$_{101-225}$, we confirmed complex formation (Fig 2B). Further characterisation using SEC-MALS showed that CAL1$_{1-50}$–CENP-A$_{101-225}$–H4 is a 1:1:1 complex with a measured MW of 39.6 ± 0.7 kDa (calculated MW 34.1 kDa) (Fig 2B).

Within CAL1, the conserved residues in α1: W22 and F29, and in α2: F43 are completely buried in the complex, so we hypothesised that these interactions are crucial for CENP-A/H4 binding (Fig 2A). To test this, we produced recombinant His-CAL1$_{1-160}$ carrying either

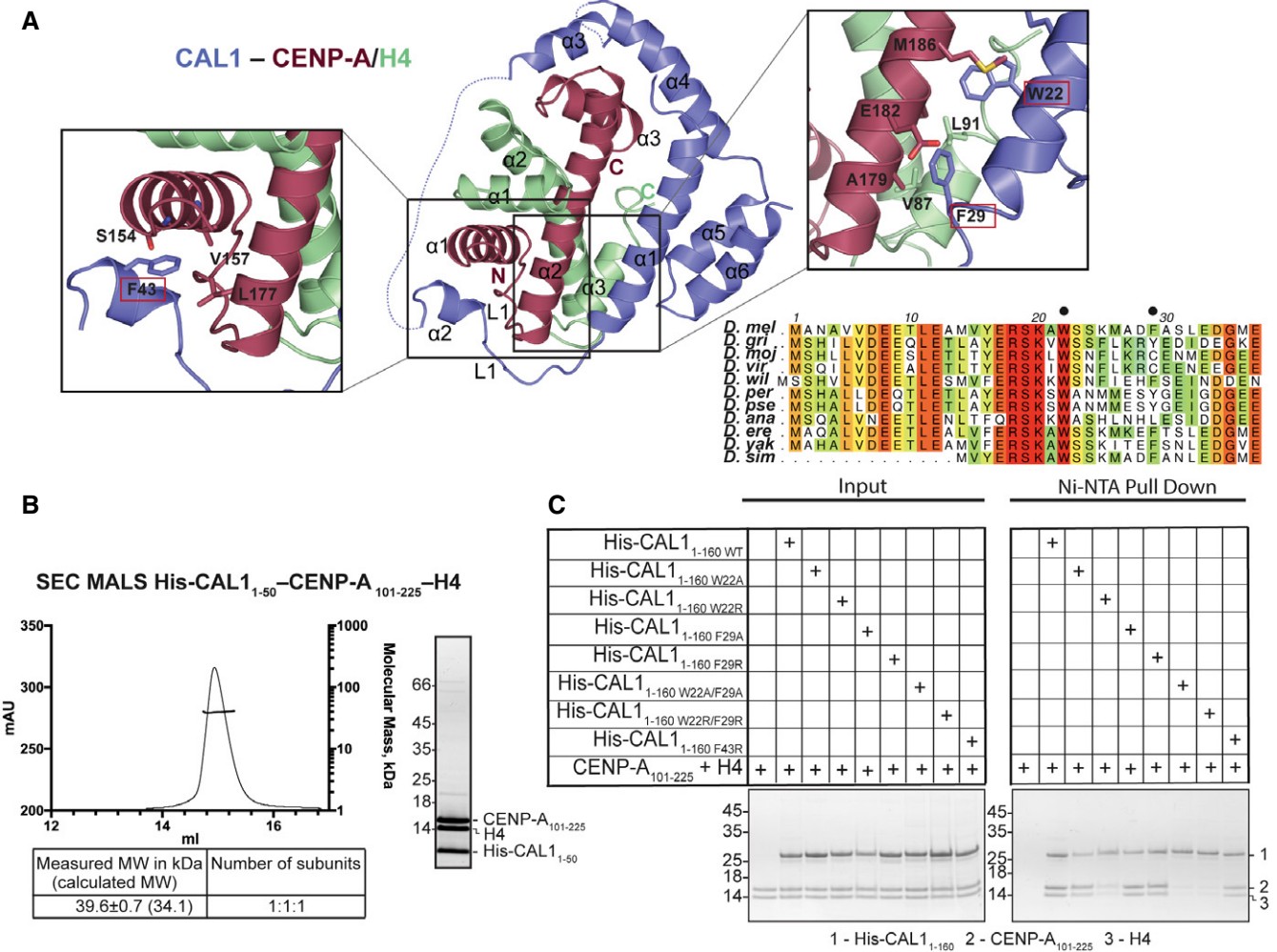

**Figure 2. Hydrophobic interactions between CAL1 α1 and CENP-A α2 are critical for CENP-A/H4 binding *in vitro*.**

A   Crystal structure of His-CAL1$_{1-160}$–CENP-A$_{144-225}$–H4 highlighting key residues involved in interaction. CAL1 is shown in blue, CENP-A in maroon and H4 in green. Multiple sequence alignment performed with MUSCLE (Madeira *et al*, 2019) showing conservation of CAL1 homologues in different fly species. Numbers correspond to *Drosophila melanogaster*. *Drosophila melanogaster* (*D. mel*), *Drosophila grimshawi* (*D. gri*), *Drosophila mojavensis* (*D. moj*), *Drosophila virilis* (*D. vir*), *Drosophila willistoni* (*D. wil*), *Drosophila persimilis* (*D. per*), *Drosophila pseudoobscura pseudoobscura* (*D. pse*), *Drosophila ananassae* (*D. ana*), *Drosophila erecta* (*D. ere*), *Drosophila yakuba* (*D. yak*) and *Drosophila simulans* (*D. sim*).

B   SEC-MALS of His-CAL1$_{1-50}$–CENP-A$_{101-225}$–H4. Absorption at 280 nm (mAU, left *y*-axis) and molecular mass (kDa, right *y*-axis) are plotted against elution volume (ml, *x*-axis). Measured molecular weight (MW) and the calculated subunit stoichiometry based on the predicted MW of different subunit compositions. Samples were analysed using a Superdex 200 increase in 50 mM HEPES pH8.0, 2 M NaCl and 1 mM TCEP.

C   Ni-NTA pull-down of His-CAL1$_{1-160\ WT}$ and indicated mutants with CENP-A$_{101-225}$–H4. SDS–PAGE shows input and protein bound to beads. Quantifications shown in Fig EV4D.

F43R, W22A, F29A, W22R, F29R, W22A/F29A or W22R/F29R mutations and tested their ability to interact with CENP-A/H4 complex in a nickel-NTA pull-down assay. His-CAL1$_{1-160}$ was mixed with molar excess of CENP-A/H4 complex. His-CAL1$_{1-160}$, and any proteins bound to it, was captured with nickel-NTA resin and subsequently analysed by SDS–PAGE. While the F43R mutation had small effect on CAL1$_{1-160}$ binding, the W22R and W22/F29 double mutations significantly reduced the ability of CAL1$_{1-160}$ to capture CENP-A/H4 compared with the WT protein (Figs 2C and EV4D left panel).

To validate the requirement of these interactions in cells, we expressed CENP-A-GFP-LacI in U2OS cells containing a synthetic array with a LacO sequence integrated in a chromosome arm

(Janicki *et al*, 2004) and analysed its ability to recruit CAL1-V5 (Roure *et al*, 2019). When CENP-A-GFP-LacI was tethered to the LacO site, CAL1$_{WT}$ was efficiently recruited (Fig 3A). Consistent with our *in vitro* binding assay, CENP-A-GFP-LacI recruited CAL1$_{F43R}$ threefold less efficiently when compared to CAL1$_{WT}$. CAL1$_{W22/F29A}$ and CAL1$_{W22/F29R}$ showed an even stronger reduction in their ability to associate with CENP-A (Fig 3A).

We also tested the recruitment of CAL1-V5 WT and mutants by co-transfecting them with CENP-A-GFP-LacI into a physiologically related *Drosophila* Schneider S2 cells containing a LacO array (Fig 3B). In agreement with the interaction studies in U2OS cells, association of CAL1$_{F43R}$ and CAL1$_{W22/F29R}$ with CENP-A was

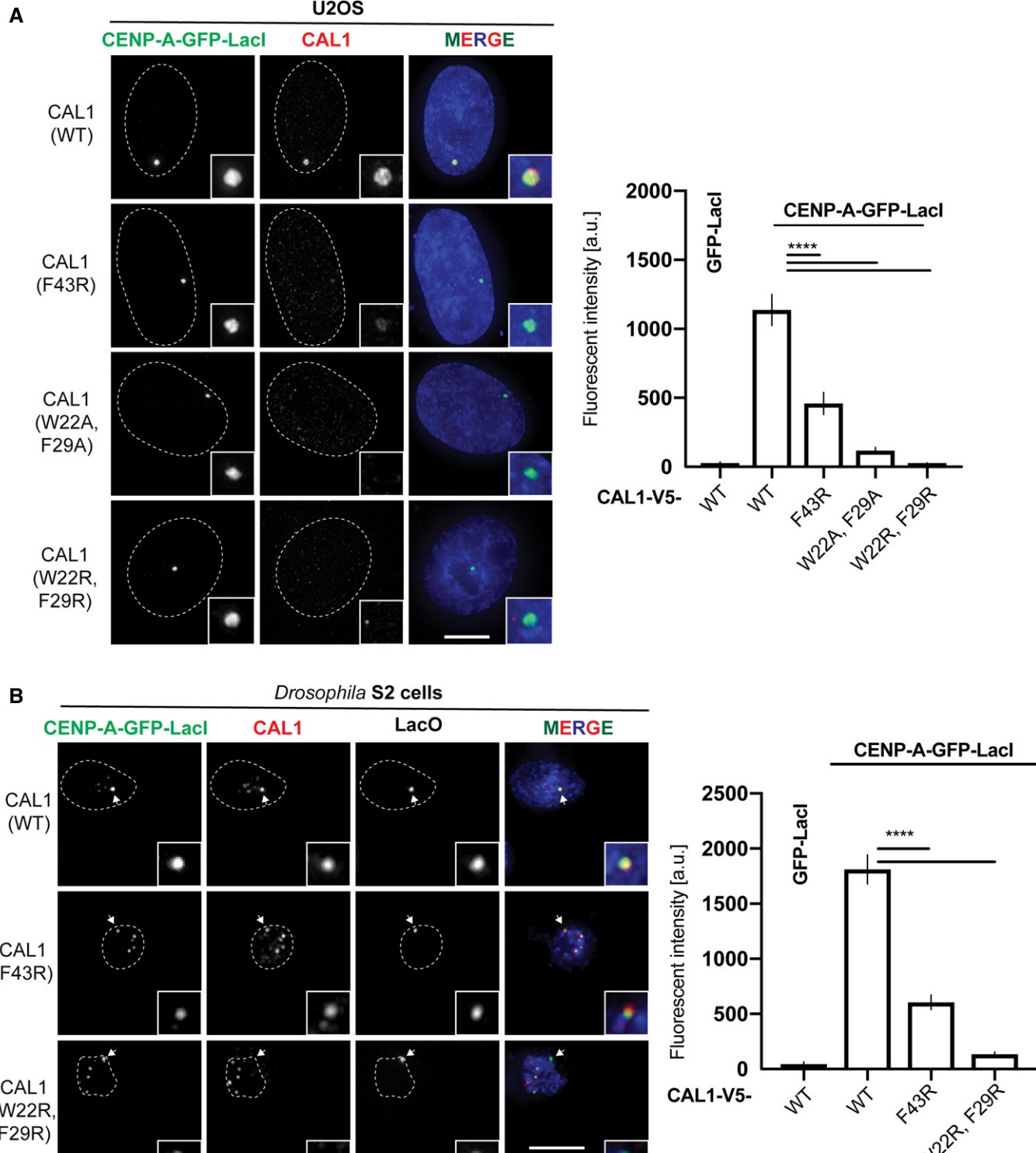

**Figure 3. Hydrophobic interactions between CAL1 α1 and CENP-A α2 are critical for CENP-A/H4 binding in cells.**

A  Representative fluorescence images and quantification of tethering assays. U2OS cells containing a LacO array were co-transfected with CENP-A-GFP-LacI with CAL1$_{WT}$-V5 and also with CAL1-V5 carrying point mutations. Scale bar: 10 μm ($n$ = 2 experiments).

B  Representative fluorescence images and quantification of *in vivo* tethering assays. *Drosophila* Schneider S2 cells containing a LacO array were co-transfected with CENP-A-GFP-LacI with CAL1$_{WT}$-V5 and also with CAL1-V5 carrying point mutations. Arrows point to the LacO site. Scale bar: is 5 μm ($n$ = 3 experiments).

Data information: In (A), data presented as mean ± SEM of 2 experiments, $n \geq 20$ cells per experiment. *P*-values were calculated using a Mann–Whitney test. In (B), data presented as mean ± SEM of 3 experiments, $n \geq 45$ cells per experiment, *P*-values were calculated using a Mann–Whitney test (****$P < 0.0001$).

significantly reduced at the tethering site (Fig 3B). Overall, *in vitro* binding assays together with interaction studies in cells indicate that the interactions mediated by W22 and F29 are crucial for the recognition of CENP-A/H4 by CAL1.

### CAL1 uses conserved and adaptive interactions to recognise *Drosophila* CENP-A/H4

Structural superposition of CAL1–CENP-A/H4 onto its respective human and *Kluyveromyces lactis* structures, HJURP–CENP-A/H4 (PDB: 3R45) (Hu *et al*, 2011) and Scm3–CENP-A/H4 (PDB: 2YFV) (Cho & Harrison, 2011), showed that CAL1 employs a broadly similar mode of CENP-A recognition with a few striking differences (Fig 4A). All CENP-A chaperones compared here use their α1 helix to interact with α2 of CENP-A in an anti-parallel fashion, occluding the tetramerisation of CENP-A/H4 heterodimers. However, in CAL1 the upstream segment of α1 swings away from CENP-A as compared with its counterpart in HJURP and Scm3. Structural superposition-based sequence alignments showed a key amino acid variation in *dm* CENP-A at position 186 as compared with human and yeast CENP-A: Ala is replaced with Met, an amino acid with a long side chain, which appears to push CAL1 α1 away from it (Fig 4B). This apparent weakening of CAL1 α1–CENP-A α2 interaction is likely to be compensated by CAL1 α5 and α6 which together restrains the position of α1 helix by forming a helical bundle. Our efforts to measure the binding strengths of CAL1 with (CAL1$_{1–160}$) and without α helical elements (CAL1$_{1–50}$) to bind CENP-A/H4 did not show a noticeable difference under the conditions (buffer containing at least 1M NaCl) needed for CENP-A/H4 solubility. We speculate that in a cellular context post-translational regulation such as phosphorylation or/and other intermolecular interaction involving the downstream helical segments of CAL1 might modulate CENP-A/H4 binding dynamics required for correct CENP-A recruitment at centromeres. This may be a possible explanation for why CAL1$_{1–50}$ is not sufficient for CENP-A recruitment in cells (Chen *et al*, 2014). Notably, loop L1 of both CAL1 and HJURP interacts with CENP-A L1 through main chain hydrogen bonding interactions. However, the secondary structural element downstream of L1 that interacts with the hydrophobic groove formed by CENP-A α1 and α2 is a three stranded β sheet in HJURP, while it is an α helix in CAL1. Strikingly, unlike other histone chaperones, CAL1 shields CENP-A α3 through downstream α helical elements (Figs 1 and 4). This intermolecular interaction appears to be critical for CENP-A recognition as a CENP-A chimera where CENP-A α3 was replaced with histone H3 α3 failed to associate with centromeres (Roure *et al*, 2019).

### CAL1 recognises amino acid variations unique to CENP-A

The histone fold domain of CENP-A and histone H3 shares 31% sequence identity. To understand how CAL1 differentiates CENP-A from histone H3, we looked for conserved CENP-A-specific amino acid variations in several *Drosophila* species and compared these variations against *dm* histone H3 (Fig 4C). This analysis together with the structural superposition of CENP-A onto histone H3 revealed several residues unique to CENP-A within the CAL1 binding region potentially responsible for CENP-A specificity: Ser154, Met186 and Gln190. The equivalent residues in histone H3 are Gln,

Ala and Gly, respectively. To evaluate whether any of these specific amino acid variations are responsible for providing CENP-A specificity, we made several recombinant CENP-A mutants where these residues are mutated to corresponding histone H3 residues (CENP-A$_{101–225\ S154Q}$, CENP-A$_{101–225\ M186A}$ and CENP-A$_{101–225\ Q190G}$) and tested their ability to interact with His-CAL1$_{1–160}$ in a nickel-NTA pull-down assay (Figs 4D and EV4D right panel). While His-CAL1$_{1–160}$ interacted with CENP-A mutants harbouring single "histone H3-like" mutations as efficiently as it does the WT CENP-A, combining three "histone H3-like" mutations resulted in a significant reduction in CAL1 binding (Figs 4D and EV4D right panel). This suggests that CAL1 achieves CENP-A specificity by recognising multiple CENP-A-specific amino acid variations.

### CAL1 chaperones CENP-A/H4 by shielding protein/DNA interaction surfaces crucial for nucleosome assembly

Histone chaperones are key regulators of nucleosome assembly. This function is achieved by ensuring the correct histone incorporation in a spatio-temporally controlled manner. To understand how CAL1 exerts its CENP-A chaperone function, we performed structural superposition of CAL1–CENP-A/H4 complex onto the crystal structure of nucleosome core particle (PDB: 2PYO) (Clapier *et al*, 2008). This revealed that CAL1 shields the CENP-A/H4 regions critical for nucleosome assembly at: (i) the CENP-A/H4 tetramerisation interface, (ii) the H2A/H2B binding region and (iii) the DNA-binding region (Fig 5). CENP-A/H4 tetramerisation is thought to be the very first step in the nucleosome assembly pathway, followed by the wrapping of DNA by the CENP-A/H4 heterotetramer and incorporation of H2A/H2B heterodimers (Hammond *et al*, 2017). Thus, the CAL1 bound form of CENP-A/H4 cannot be incorporated into the nucleosome, inhibiting any unwarranted incorporation of CENP-A.

### CENP-C binds CAL1 via its C-terminal cupin domain

We next aimed to understand the structural basis for the centromere targeting of the CAL1 bound pre-nucleosomal CENP-A/H4 heterodimer. Previous studies have shown that CAL1 and CENP-C can directly interact with each other through their C-terminal regions, CAL1$_{699–979}$ and CENP-C$_{1009–1411}$ (Fig 6A), respectively (Schittenhelm *et al*, 2010). However, efforts to purify these recombinant proteins were not successful as they were prone to degradation. Based on secondary structure prediction and sequence conservation analysis, we designed shorter constructs, CAL1$_{841–979}$ and CENP-C$_{1264–1411}$. This CENP-C fragment contains an evolutionarily conserved cupin domain. Reconstitution of CAL1–CENP-C complex using individually purified His-SUMO-CAL1$_{841–979}$ and His-CENP-C$_{1264–1411}$ showed clear complex formation (Fig 6B): His-SUMO-CAL1$_{841–979}$ eluted at a volume of 10.38 ml, His-CENP-C$_{1264–1411}$ 10.54 ml, while the complex eluted at 9.63 ml.

### Overall structure of the CENP-C cupin domain

A well-conserved structural feature of CENP-C among different species is the presence of a C-terminal cupin domain. Previous structural characterisation of the cupin domain of Mif2p, the budding yeast orthologue of human CENP-C, showed that it forms a dimer (Cohen *et al*, 2008). Although CENP-Cs across species contain

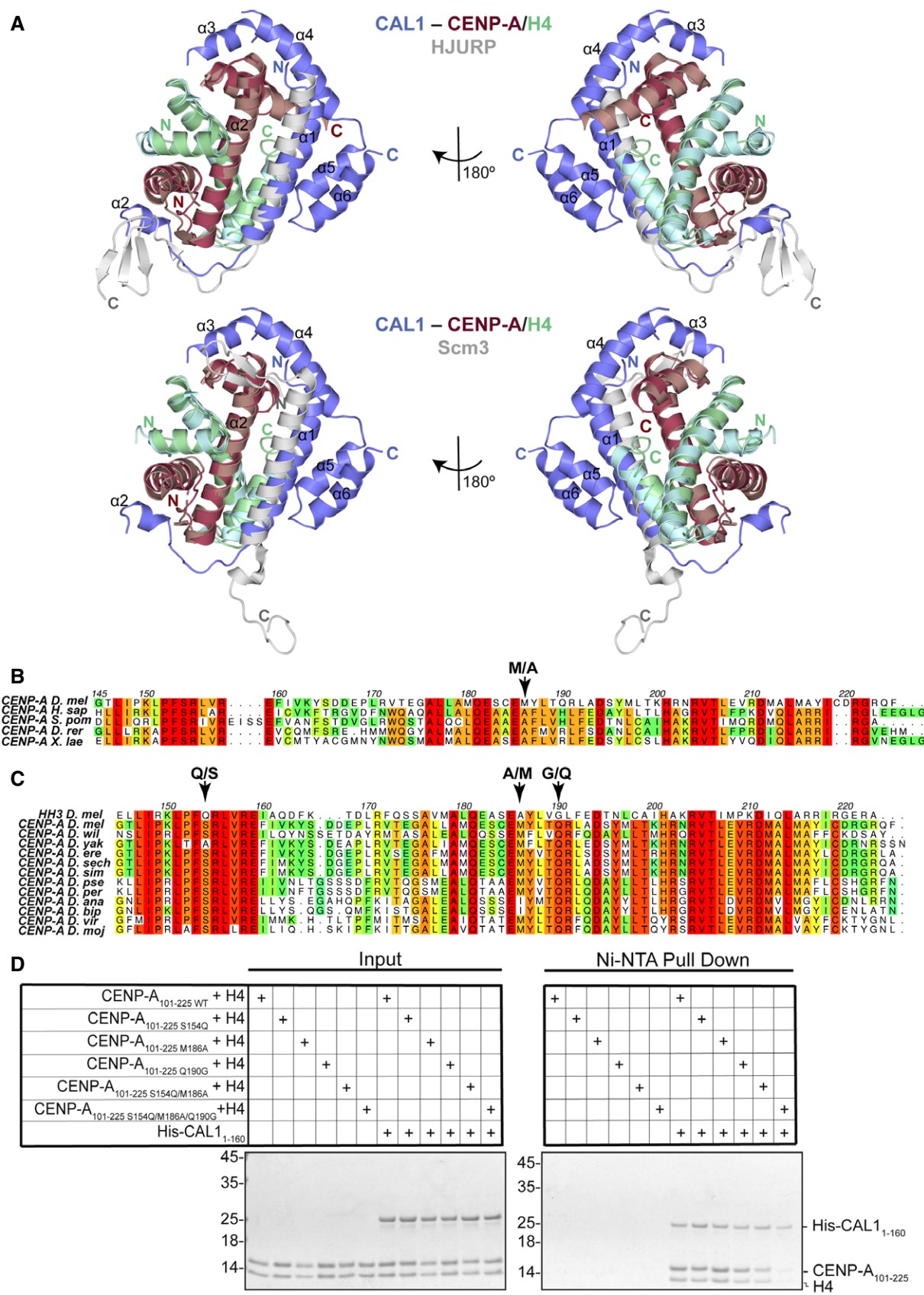

**Figure 4.**

a C-terminal cupin domain, these appear to show striking amino acid variations. Pairwise sequence alignments of *dm* CENP-C cupin domain against its budding yeast counterpart showed 11% sequence identity and 18% sequence identity against the human counterpart. Crystallisation trials carried out with CENP-C$_{1264-1411}$ alone and in complex with CAL1 produced diffraction quality crystals which diffracted X-rays to about 1.8 and 2.3 Å, respectively (Table 1).

The CENP-C$_{1264-1411}$ structure was determined by molecular replacement using the crystal structure of budding yeast Mif2p cupin domain (PDB: 2VPV) (Cohen *et al*, 2008). The twofold axis of the CENP-C$_{1264-1411}$ dimer was aligned with the crystallographic twofold axis. Consequently, just one molecule was present in the asymmetric unit (Fig 6C). As expected, CENP-C$_{1264-1411}$ domain forms a cupin fold almost entirely made of β strands forming a β-barrel with a helix preceding the cupin domain. The β strands assemble into two β sheets: a six-stranded (β1-β2-β3-β10-β5-β8) and a four-stranded (β4-β9-β6-β7) (Fig 6C). The β1 of the six-stranded β sheet is connected to the preceding α1 (spanning aa residues 1,276–1,288) with a long loop (aa residues 1,289–1,313) containing two short α helical segments. Dimerisation of CENP-C cupin domain is mediated by a back-to-back arrangement of six-stranded β-sheets. In this arrangement, the loop connecting the N-terminal α helix (α1) to β1 crosses over to its dimeric counterpart resulting in a "roof"-like positioning of α-helices on top of the β barrels. The surface area buried at the dimerisation interface is 1,706 Å$^2$ which is about 50% of the total solvent accessible surface area. The interactions stabilising the dimerisation are predominantly hydrophobic involving residues L1283, W1286, L1287, L1312, L1314, Y1325, Y1335, M1407 and L1357 (Fig 6D). Among these residues, L1357 and M1407 are centrally located and juxtaposed within the hydrophobic core. This led us to hypothesise that these residues may be critical for the assembly of cupin dimer. To test this, we generated a mutant where L1357 and M1407 were mutated to glutamic acids (CENP-C$_{1264-1411\ L1357E/M1407E}$) and analysed their oligomeric structure by measuring the MW using SEC-MALS (Fig 6E). While the measured MW of CENP-C$_{1264-1411}$ agreed with the calculated MW of a dimer, the corresponding value for the His-CENP-C$_{1264-1411\ L1357E/M1407E}$ revealed that it was a monomer (measured MW 20.2 ± 0.4 kDa and calculated MW 19.2 kDa) (Fig 6E).

Structural comparison of *dm* and budding yeast CENP-C cupin domains showed that although these domains share only weak similarity at the amino acid sequence level (21%), the overall fold conferring the β barrel structure is conserved. However, two loop regions (*dm* CENP-C 1,324–1,333 and 1,368–1,376) show striking conformational variation as compared with their equivalent regions in budding yeast CENP-C, Mif2p (Fig EV5A).

During the preparation of this manuscript, work from elsewhere reported a crystal structure of a slightly longer fragment of dm CENP-C spanning aa 1,190–1,411 (PDB: 6O2K). The structure reported here and PDB: 6O2K are nearly identical and superpose well with an RMSD of 0.27 Å (Chik *et al*, 2019; Fig EV5B).

**Structural basis for CAL1 recognition by CENP-C**

The structure of CENP-C$_{1264-1411}$ bound to CAL1$_{841-979}$ was determined by molecular replacement using the CENP-C$_{1264-1411}$ structure reported here as a search model. The final model was refined to $R$ and $R_{free}$ factors of 23.7 and 26.6%, respectively, and included CENP-C residues 1,303–1,411 and CAL1 residues 890–913 (Fig 7A). This suggests that CAL1 residues preceding and following the region 890–913 are flexible and are not stabilised by CENP-C. While CAL1 residues 890–893 form a β-strand, residues 894–913 form a highly basic α helix (calculated pI of 10.57). CENP-C binds CAL1 using a cradle-shaped surface formed by loops L1, L2 and L3 and β-strands β1 and β2. The calculated electrostatic surface properties show that CAL1 binding involves a surface suitable for both electrostatic and hydrophobic interactions (Fig 7A). CAL1 residues 890–893, which form a β-strand, interact with β1 of CENP-C cupin domain running parallel to it and as a consequence extend the β sheet involved in cupin dimerisation. The CAL1 α helix consisting of residues 894–913 makes several hydrophobic (involving L896, I900, W904 and Y908) and electrostatic (R903 and K906) interactions with a complementary hydrophobic (involving residues Y1315, V1317, Y1322 and F1323) and acidic (S1295, E1311 and N1326) region of the cradle-shaped CENP-C surface (Fig 7A–C). To evaluate the requirement of these interactions to stabilise CAL1–CENP-C binding, we mutated conserved CENP-C F1324 to Arg and CAL1 I900 to Arg and K907 and Y908 to Ala and tested the ability of these mutants to bind wild-type CAL1 and CENP-C, respectively, in separate SEC experiments (Fig 8A and B). Both His-CENP-C$_{1264-1411\ F1324R}$ and His-SUMO-CAL1$_{841-979\ I900R/K907A/Y908A}$ failed to interact with His-SUMO-CAL1$_{841-979}$ and His-CENP-C$_{1264-1411}$, respectively, and hence eluted at their original elution volumes as compared with the elution volume of the CENP-C–CAL1 complex.

We next evaluated the contribution of CENP-C and CAL1 residues identified here as critical for interaction *in vitro* in U2SO and *Drosophila* Schneider S2 cells where LacO arrays are integrated in one of the chromosome arms. Tethering GFP-LacI-CENP-C recruited CAL1-V5 to the LacO array. However, the F1324R or the L1357E/M1407E mutation in CENP-C and I900R/K907A/Y908A mutations in CAL1 are both able to inhibit interaction and reduce co-localisation at the tethering site (Fig 8D and E).

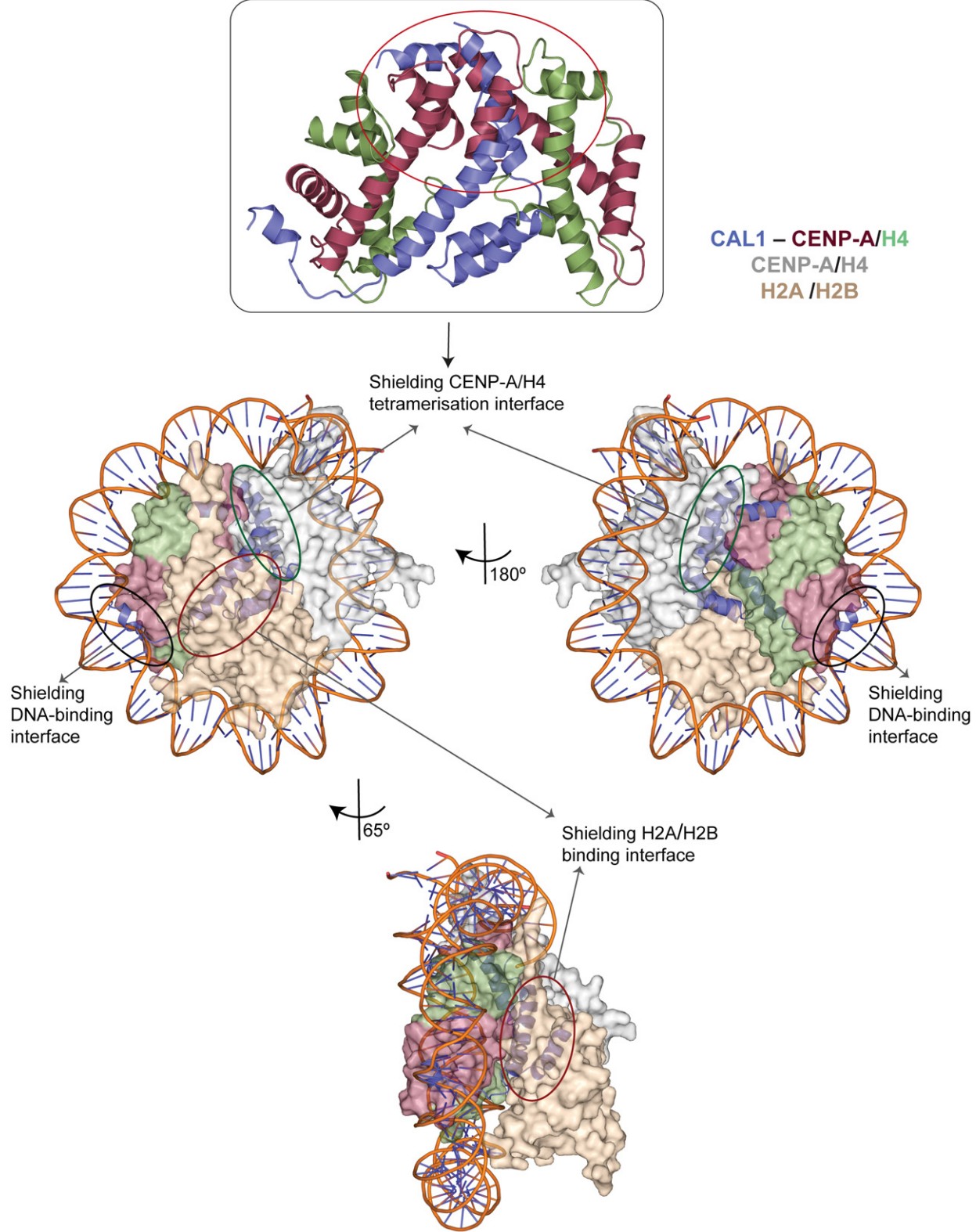

**Figure 5. CAL1 chaperones CENP-A/H4 by shielding the CENP-A/H4 tetramerisation, DNA-binding and H2A/H2B-binding interfaces.**

Structure of His-CAL1$_{1-160}$–CENP-A$_{144-225}$–H4 superimposed with modelled structure of the *dm* CENP-A nucleosome (Clapier *et al*, 2008). Multiple orientations highlighting how key CENP-A/H4 surfaces required for nucleosome assembly are shielded by CAL1-binding. CENP-A/H4 and H2A/H2B are shown in surface representation. CAL1 and DNA are shown in cartoon representation. CENP-A and H4 bound to CAL1 are shown in maroon and green, while CAL1 in blue. Top panel shows the structure of CAL1$_{1-160}$ superimposed onto the *dm* CENP-A/H4 tetramer (Clapier *et al*, 2008) to highlight steric clash.

### Dimerisation of the CENP-C cupin domain stabilises the CAL1 binding site

Previously, we showed that CENP-C dimerisation is required for CAL1 binding in cells (Roure *et al*, 2019). In the crystal structure presented here, the CAL1 binding site on CENP-C is in close proximity to the cupin dimerisation interface: the loop L1 and β-strands β1, β2 and β3 are all directly involved in stabilising the cupin dimer. This led us to hypothesise that the CAL1 binding site is stabilised in the right conformation by the dimerisation interface and hence disrupting the dimerisation interface might affect CAL1 binding. To test this, we evaluated using SEC the ability of

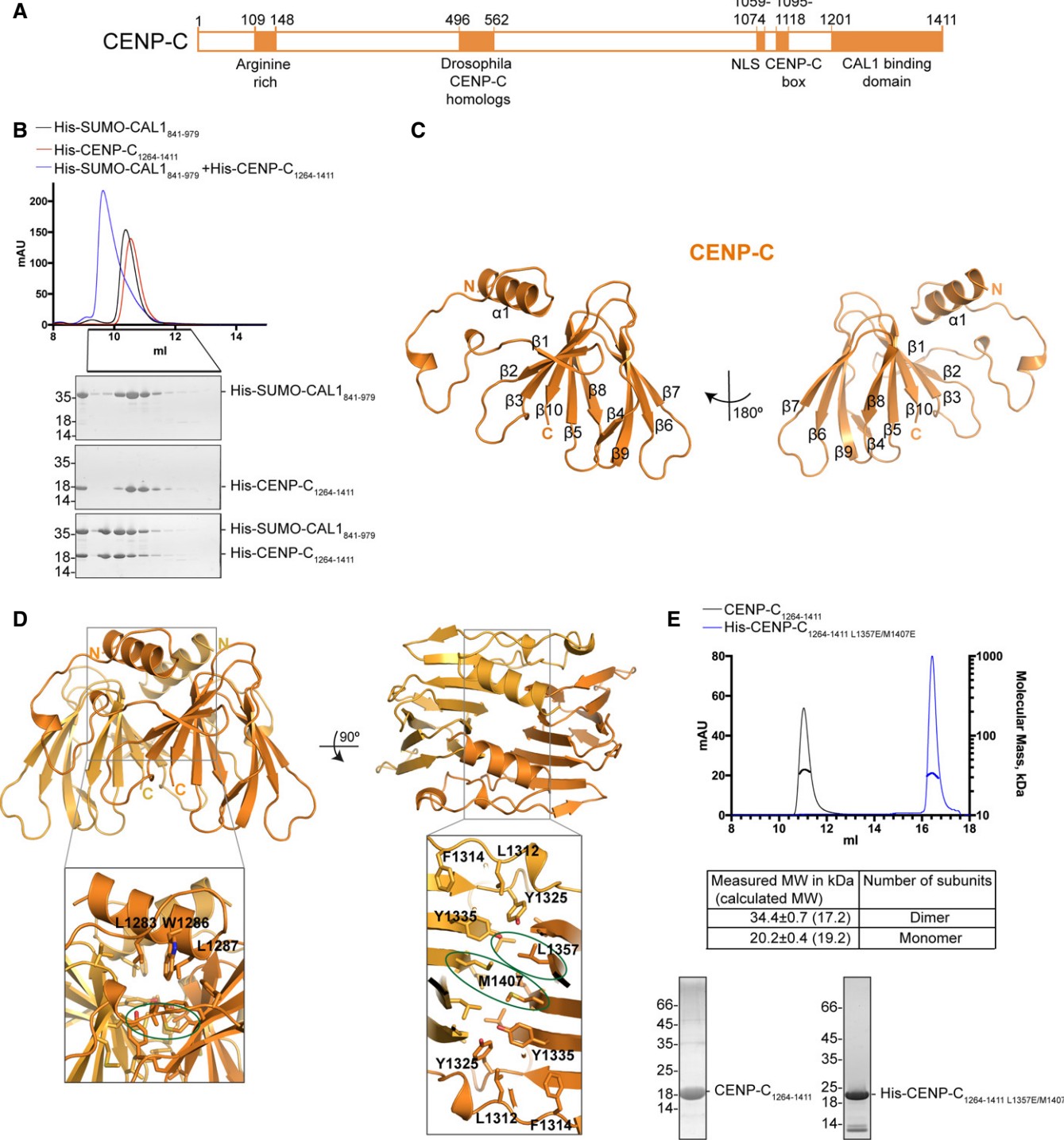

**Figure 6.**

◀

**Figure 6. CAL1 binds CENP-C by directly interacting with the evolutionarily conserved Cupin domain.**

A  Schematic representation of the structural features of CENP-C. Filled boxes represent domains.
B  SEC profile of His-SUMO-CAL1$_{841–979}$ (black), His-CENP-C$_{1264–1411}$ (red) and His-SUMO-CAL1$_{841–979}$ mixed with molar excess of His-CENP-C$_{1264–1411}$ (blue) and corresponding SDS–PAGE analysis of the fractions. Samples were analysed using Superdex 75 increase 10/300 in 20 mM Tris–HCl pH 8.0, 100 mM NaCl and 2 mM DTT.
C  Crystal structure of CENP-C cupin domain determined at 1.7 Å resolution.
D  Overall structure of CENP-C cupin domain dimer. Amino acid residues involved in dimerisation are highlighted in zoomed in panels. Residues mutated to disrupt dimerisation are circled.
E  SEC-MALS analysis of CENP-C$_{1264–1411}$ (black) and His-CENP-C$_{1264–1411\ L1357E/M1407E}$ (blue). Absorption at 280 nm (mAU, left $y$-axis) and molecular mass (kDa, right $y$-axis) are plotted against elution volume (ml, $x$-axis). Measured MW and the calculated subunit stoichiometry based on the predicted MW of different subunit compositions. Samples were analysed using either a Superdex 75 or a Superdex 200 increase 10/300 in 50 mM HEPES pH 8.0, 100 mM or 300 mM NaCl and 1 mM TCEP.

His-CENP-C$_{1264–1411\ L1357E/M1407E}$, which we have shown here is not capable of forming a dimer (Fig 6E), to bind CAL1. When His-SUMO-CAL1$_{841–979}$ was mixed with 1.2 molar excess of His-CENP-C$_{1264–1411\ L1357E/M1407E}$ and subjected to SEC analysis, they did not interact with each other and eluted separately at elution volumes 10.4 and 11.6 ml, respectively (Fig 8C). Consistent with these *in vitro* data, GFP-LacI-CENP-C tethered to the LacO site in U2OS and *Drosophila* S2 cells recruited CAL1 robustly, while the GFP-LacI-CENP-C$_{L1357E/M1407E}$ failed to do so (Fig 8D and E). These observations together demonstrate that the CENP-C dimerisation-mediated stabilisation of CAL1 binding site is an essential requirement for CENP-C association of CAL1.

### The CENP-C cupin dimer binds just one CAL1 molecule

Although CAL1 binding by CENP-C involves just a cupin monomer, only one of the two cupin monomers was observed to interact with CAL1, while the equivalent CAL1 binding site of the dimeric counterpart was empty in the crystal structure. We speculate that the other binding site might be sterically hindered by the remaining residues of CAL1 not seen in the crystal structure, thus not allowing a second monomer of CAL1 to bind. This agrees with our previous observation that CAL1$_{841–979}$ and CENP-C$_{1264–1411}$ form a 1:2 complex in solution as estimated using the mass spectrometry derived iBAC peptide ratio and SEC-MALS (Roure *et al*, 2019). To confirm the subunit stoichiometry of CAL1-CENP-C complex unambiguously, we measured the molecular mass of CAL1$_{841–979}$–CENP-C$_{1264–1411}$ complex using analytical ultracentrifugation (AUC) (Fig 9A). First, the individual components of the complex were characterised by both sedimentation velocity (SV) and sedimentation equilibrium (SE), the data from which (Fig EV5C) demonstrate that CAL1$_{841–979}$ is monomeric with a very weak tendency to self-associate, while CENP-C$_{1264–1411}$ is a dimer (95% confidence intervals for MW are 17.453 and 21.086 for CAL1 and 33.901 and 36.311 for CENP-C). Next, samples comprising a SEC purified untagged complex were analysed. Both the mass and sedimentation coefficient are consistent with a 2:1 complex, but not with a 2:2 complex (95% confidence intervals for MW are 38.896 and 58.881 for the complex). Thus, the AUC together with the crystal structure shows that CENP-C cupin dimer binds just one copy of CAL1 at any given time.

### CAL1 can bind CENP-A/H4 and CENP-C at the same time

To gain a better understanding of the mechanism of CENP-A deposition, we wanted to establish whether CAL1 could bind both CENP-A/H4 and CENP-C simultaneously. However, we have not been able to generate recombinant full-length CAL1 using bacteria or insect cells. Since we have already established the regions needed to bind CENP-A/H4 and CENP-C, we generated an engineered version of CAL1 with 1–160 and 841–979 connected by a flexible GSSGGSSG linker. This was expressed and refolded with CENP-A$_{101–225}$ and H4 in a similar manner to CAL1$_{1–160}$. The resulting folded complex was analysed by SEC on its own and mixed with 1.2 molar excess of His-CENP-C$_{1264–1411}$ (Fig 9B). This revealed a clear shift in the elution profile of His-CAL1$_{1–160-LL-841–979}$/CENP-A$_{101–225}$/H4 when bound to His-CENP-C$_{1264–1411}$ (14.24 ml) compared to His-CAL1$_{1–160-LL-841–979}$/CENP-A$_{101–225}$/H4 (14.62 ml) and His-CENP-C$_{1264–1411}$ (16.05 ml) on their own (Fig 9B).

## Discussion

Understanding the molecular details of how organisms maintain their centromere identity has been of great importance to biologists as loss of centromeres or establishment of new centromeres at non-centromeric locus (neocentromeres) results in genome instability, often leading to cell death. To maintain centromere identity defined by the enrichment of CENP-A containing nucleosome, the CENP-A-specific chaperone (HJURP in humans and Scm3 in yeast) escorts CENP-A until its incorporation into the centromeric chromatin (Dunleavy *et al*, 2009; Foltz *et al*, 2009; Pidoux *et al*, 2009). Correct spatio-temporal regulation of this process is achieved by the Mis18 complex in humans and fission yeast (Hayashi *et al*, 2004; Fujita *et al*, 2007; Foltz *et al*, 2009; McKinley & Cheeseman, 2014; Pan *et al*, 2017; Spiller *et al*, 2017). Despite the essential requirement of CENP-A deposition at centromeres, the pathways and the molecular players regulating this process show significant variations across organisms (Zasadzinska & Foltz, 2017). This suggests that these organisms have evolved to employ unique strategies to establish and maintain centromeric chromatin.

*Drosophila* is a remarkable model organism to study centromere inheritance as it lacks direct homologues of either HJURP and Scm3 or the Mis18 complex. Instead, it maintains centromere identity using just CAL1. CAL1 does not share obvious sequence similarity with Scm3 or HJURP and does not appear to share common ancestry with these chaperones (Sanchez-Pulido *et al*, 2009; Phansalkar *et al*, 2012; Rosin & Mellone, 2016). Our structural analysis presented here shows that although CAL1 appears to have evolved independently of Scm3 and HJURP, it employs evolutionarily conserved and adaptive structural principles to bind CENP-A.

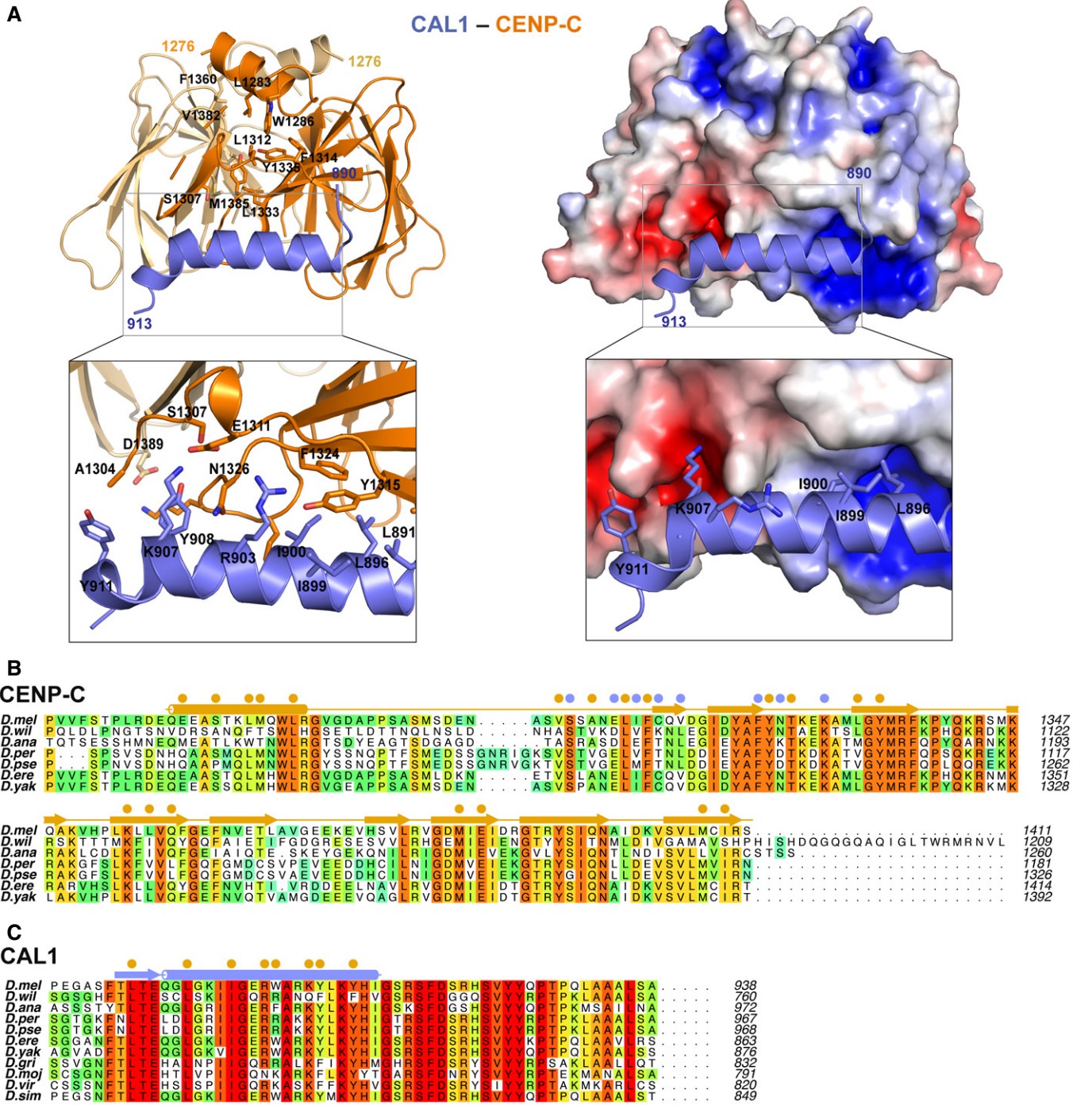

**Figure 7. Structural basis for CAL1 recognition by CENP-C cupin domain.**

A    (left panel) Crystal structure of CAL1 (shown in blue) bound CENP-C cupin domain (shown in orange) determined at 2.4 Å resolution shown in cartoon representation. (right panel) CENP-C cupin domain is shown in surface representation coloured based on electrostatic surface potential calculated using APBS. Zoomed in views highlight residues involved in interaction.

B, C    Multiple sequence alignment performed with MUSCLE (Madeira *et al*, 2019) (B) showing amino acid conservations of CENP-C homologues in different fly species. Residues involved in CENP-C cupin dimerisation and CAL1 binding are highlighted with filled orange and blue circles, respectively, (C) showing conservations of C-terminus of CAL1 across its homologues in different fly species. Orange filled circles highlight the residues involved in CENP-C binding. *Drosophila melanogaster (D. mel), Drosophila willistoni (D. wil), Drosophila ananassae (D. ana), Drosophila persimilis (D. per), Drosophila pseudoobscura pseudoobscura (D. pse), Drosophila erecta (D. ere), Drosophila yakuba (D. yak), Drosophila grimshawi (D. gri), Drosophila mojavensis (D. moj), Drosophila virilis (D. vir), and Drosophila simulans (D. sim).*

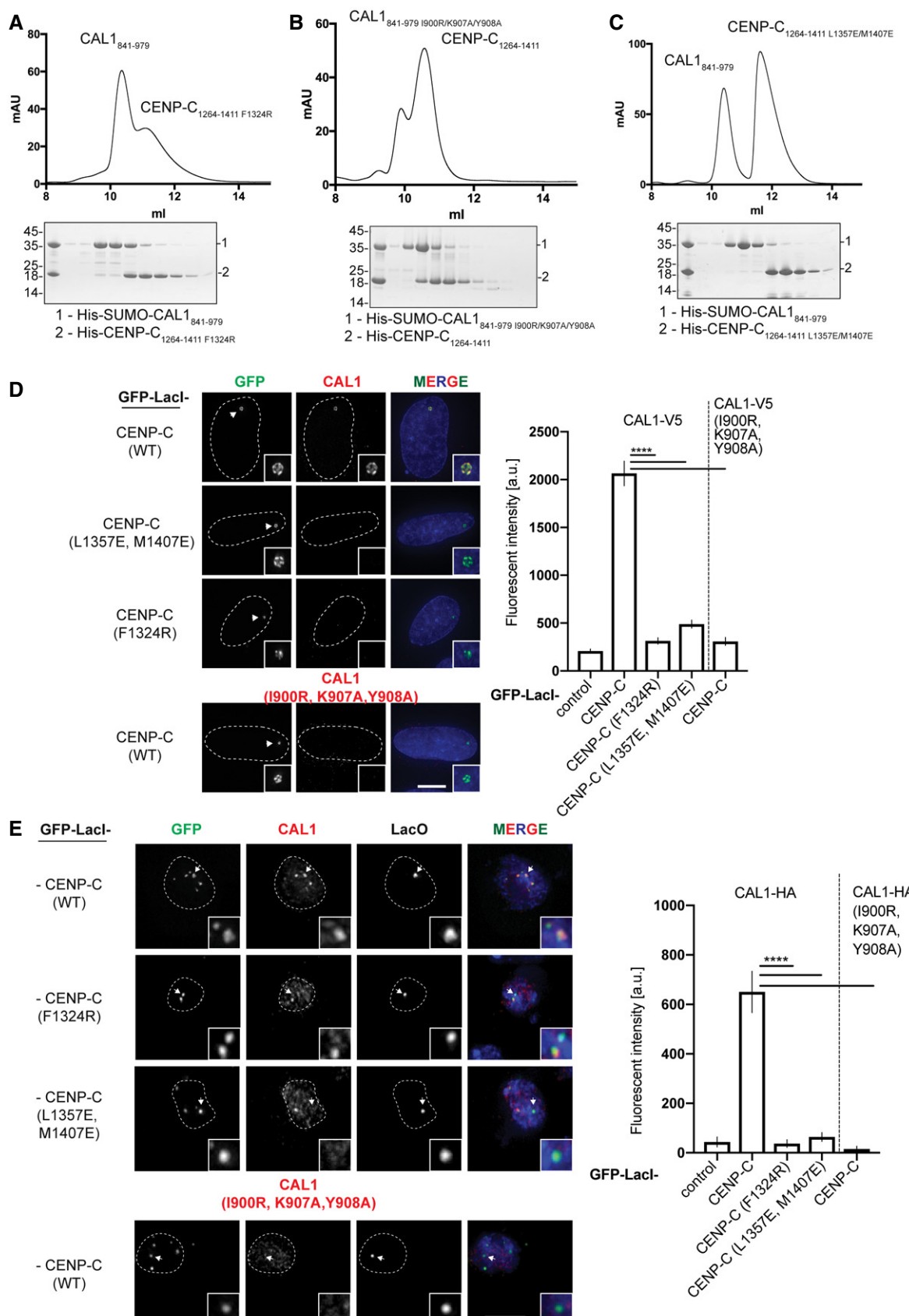

Figure 8.

**Figure 8. CENP-C cupin dimerisation is critical for CAL1 binding.**

A–C SEC analysis of (A) His-CENP-C$_{1264–1411}$ $_{F1324R}$ and His-SUMO-CAL1$_{841–979,}$ (B) His-CENP-C$_{1264–1411}$ and His-SUMO-CAL1$_{841–979}$ $_{I900R/K907A/Y908A}$ and (C) His-CENP-C$_{1264–1411}$ $_{L1357E/M1407E}$ and His-SUMO-CAL1$_{841–979}$. Samples were analysed using a Superdex 75 in 20 mM Tris–HCl pH 8.0, 0.1 M NaCl and 2 mM DTT. Corresponding fractions shown by SDS–PAGE with Coomassie stain underneath.

D Representative IF images and quantification of tethering assays. U2OS cells containing a LacO array were transfected with GFP-LacI-CENP-C with CAL1-V5 to assess interaction. CENP-C mutants F1324R, L1357E/M1407E and CAL1 mutant I900R/K907A/Y908A were tested in each construct separately. Scale bar: 10 μm (*n* = 4 experiments except F1324 where *n* = 3 experiments).

E Representative IF images and quantification of *in vivo* tethering assays. *Drosophila* Schneider S2 cells containing a LacO array were transfected with GFP-LacI-CENP-C and CAL1-HA to assess interaction. Arrows point to the LacO site. Scale bar is 5 μm (*n* = 3 experiment, except CAL1 I900R/K907A/Y908A where *n* = 2 experiments).

Data information: In (D), data presented as mean ± SEM of 3 or 4 experiments, *n* ≥ 22 cells per experiment. *P*-values were calculated using an Mann–Whitney test. In (E), data presented as mean ± SEM of 3 experiments, except for CAL1 mutant I900R/K907A/Y908A which was 2 experiments. *n* ≥ 44 cells per experiment, *P*-values were calculated using Mann–Whitney test (****$P$ < 0.0001).

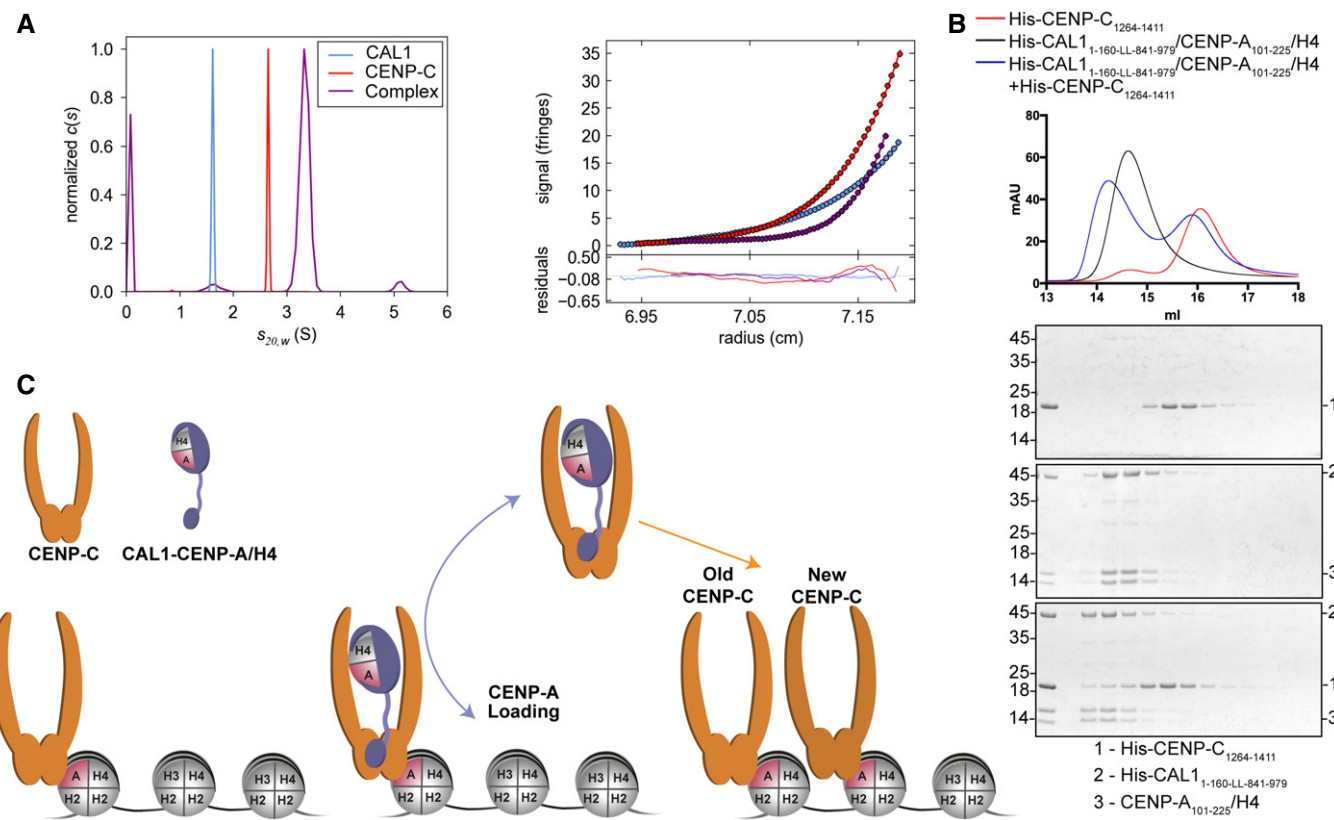

**Figure 9. CAL1 can associate with CENP-A/H4 heterodimer and CENP-C simultaneously.**

A (left panel) Normalised sedimentation coefficient distribution (c(s)) for CAL1$_{841–979}$ (CAL1, blue), CENP-C$_{1264–1411}$ (CENP-C, red) and their equimolar mix (complex, purple) all at 10 mg/ml, demonstrating a significant increase in $s^0_{20,w}$ consistent with the formation of a 2:1 complex. (right panel) Typical sedimentation equilibrium data for CAL1$_{841–979}$ (CAL1, blue), CENP-C$_{1264–1411}$ (CENP-C, red) and their equimolar mix (complex, purple) all at 10 mg/ml, demonstrating a significant increase in mass, consistent with the formation of a 2:1 complex. The data were fit with a single species model yielding masses of 20,253, 29,216 and 49,539 g/mol. The values reported in Fig EV5C are based on data acquired for a range of concentrations.

B SEC profile of His-CENP-C$_{1264–1411}$ (red), His-CAL1$_{1–160-LL-841–979}$/CENP-A$_{101–225}$/H4 (black) and His- His-CAL1$_{1–160-LL-841–979}$/CENP-A$_{101–225}$/H4 mixed with molar excess of His-CENP-C$_{1264–1411}$ (blue) and corresponding SDS–PAGE analysis of the fractions. Samples were analysed using Superdex 200 increase 10/300 in 20 mM Tris–HCl pH 8.0, 100 mM NaCl and 2 mM DTT.

C Schematic model of CAL1-mediated loading of CENP-A/H4 and CENP-C binding at centromeres.

Recognition of CENP-A L1 and α2 by the N-terminal 50 aa of CAL1 is similar to that of Scm3 and HJURP. Despite this, CAL1 is also distinctly dissimilar from Scm3 and HJURP as residues downstream of the N-terminal 50 aa wrap around CENP-A/H4 making additional contacts with CENP-A α3 and CAL1 itself. These interactions appear to be crucial for CENP-A deposition as the N-terminal 50 aa of CAL1 were not sufficient to recruit CENP-A to centromeres in cells (Chen *et al*, 2014). Notably, unlike the human CENP-A, the centromere

targeting domain of *Drosophila* CENP-A includes α3 as L1 and α2 were not sufficient to target CENP-A (Roure *et al*, 2019). We note that while HJURP and Scm3 fragments used for structure analysis are shorter than the CAL1 fragment used here, secondary structure prediction analysis suggests a lack of a similar α-helical segments downstream of HJURP/Scm3 α1 that stacks against CENP-A α2.

When compared to the available "histone variant"—chaperone complex crystal structures, the overall mode of CENP-A/H4 recognition by CAL1 appears to be novel as it is the only one which wraps around CENP-A/H4 through multiple CENP-A and H4 contacts resulting in the shielding of CENP-A/H4 surfaces involved in CENP-A/H4 tetramerisation, DNA binding and H2A/H2B binding—all critical for nucleosome assembly. This is in agreement with the observation that CAL1 cannot directly interact with the CENP-A nucleosome (Roure *et al*, 2019) and requires CENP-C to mediate the interaction with the centromeric chromatin.

In humans and fission yeast, the Mis18 complex is responsible for targeting the HJURP bound pre-nucleosomal CENP-A/H4 to the centromere by directly binding CENP-C (reviewed in Stellfox *et al*, 2012; Westhorpe & Straight, 2014; Zasadzinska & Foltz, 2017) but appears to have been lost during evolution in *Drosophila*. However, CAL1 seems to compensate for this loss by directly associating with CENP-C, which is present in most organisms with monocentric chromosomes (Drinnenberg *et al*, 2014). While there has been a suggestion that CENP-C cupin domain could be a dimer based on the crystal structure Mif2p cupin domain (Cohen *et al*, 2008), structural and functional roles of CENP-C cupin domain have remained unclear. Here we show that CAL1 associates with CENP-C by directly interacting with the cupin domain and this interaction is essential for CENP-C mediated recruitment in cells. Our structural analysis shows that the overall structure of *Drosophila* CENP-C cupin domain is similar to that of *Saccharomyces cerevisiae* (Mif2p) and *Schizosaccharomyces pombe* (Cnp3) CENP-C cupin domains, with striking differences in the mode of dimerisation (Fig EV5A and B). It is tempting to suggest that this variation is related to the ability of *dm* CENP-C to bind CAL1 as the CAL1 binding interface of CENP-C is stabilised by dimerisation. In agreement with this notion, CENP-C cupin domains of *dm* and *S. pombe* appear to use different modes of binding to CAL1 and Moa1, respectively. In the case of *S. pombe* CENP-C, Moa1 binding site is mapped to an extended site with critical residues laterally spread across the deep pocket that forms the core of the cupin domain (Chik *et al*, 2019), whereas CAL1 binds at the periphery of the equivalent pocket and extends away from the pocket and towards the dimeric interface (Fig EV5D).

Interestingly, although the *dm* CENP-C cupin dimer possesses two CAL1 binding sites, it appears to accommodate just one CAL1 at a time due to steric hindrance limiting the accessibility of the second CAL1 site. This might have broader implications for the mechanism of CENP-C binding at centromeres (Roure *et al*, 2019). In the context of the full-length proteins, CAL1 can also oligomerise via its N-terminus (Roure *et al*, 2019), leading to a scenario where a CENP-C bound CAL1 at the centromere might interact with a second CAL1 bringing another CENP-A/H4 dimer and CENP-C to facilitate CENP-A/H4 tetramer incorporation and the recruitment of CENP-C to the newly formed CENP-A nucleosome (Fig 9C). This is consistent with CENP-C targeting being reliant on CAL1 and CENP-A (Goshima *et al*, 2007; Erhardt *et al*, 2008; Schittenhelm *et al*, 2010; Roure *et al*, 2019).

In summary, our work demonstrates how *Drosophila* species elegantly compensates for the loss of HJURP or Scm3 and the Mis18 complex through CAL1, which by combining evolutionarily conserved and adaptive structural interactions escorts CENP-A/H4 to the centromere for its subsequent incorporation into the chromatin to maintain centromere identity. Moreover, this is the first study providing the structural basis for how the CENP-A deposition machinery is targeted to centromeres in any organism. Future structural studies on the Mis18 complex and its interaction with HJURP and CENP-C will provide insights into how apparently complex intermolecular interactions achieve the same objective in vertebrates and what are the species-specific functional requirements of this complexity.

# Materials and Methods

## Plasmids

Codon optimised *Drosophila melanogaster* CAL1 and CENP-C were produced as gBlocks (IDT) and used directly in ligation-independent cloning (LIC) into bacterial expression vectors. Smaller fragments were amplified using PCR and then used for LIC. CAL1$_{1–160-LL-841–979}$ was produced using homologous PCR.

All mammalian expression vectors used in this study were constructed in a pN2-CMV vector.

| GFP_LacI-tagged vectors | mRFP_LacI-tagged vectors | HA-tagged vectors |
|---|---|---|
| pMT_dCENPA_GFP_LacI | pMT_mRFP_LacI | pMT_Cal1_HA |
| pMT_GFP_LacI_dCENPC | | pMT_Cal1_W22R_F29R_HA |
| pMT_CMV_GFP_LacI_dCENPC_F1324R | | pMT_Cal1_F43R_HA |
| pMT_CMV_GFP_LacI_dCENPC_L1357E_M1407E | | |
| pMT_GFP_LacI | | |

Mutants were generated following the site-directed mutagenesis protocol using phusion ultra II. Primers used are shown in Table 2. pET3a CENP-A$_{101–225}$ was generated in (Roure *et al*, 2019). pET His6 Sumo TEV (14S Addgene plasmid # 48291) was a gift from Scott Gradia. pEC-K-3C-His was a gift from Elena Conti. pET22b H4 was a kind gift from Karolin Luger.

## Protein production

### Purification of histones
Histones were expressed and purified as described in Abad *et al* (2019) (see Appendix Supplementary Methods for details).

### Purification of CAL1 and CENP-C proteins
His-CAL1$_{1–160}$, His-CAL1$_{1–50}$ and His-CAL1$_{1–160-LL-841–979}$ were expressed using *Escherichia coli* BL21 (*DE3*) Gold (Agilent) grown in 2XTY media. His-CENP-C$_{1264–1411}$ was expressed in Rosetta (*DE3*) (Novagen) cells, using 2XTY. His-SUMO-CAL1$_{841–979}$ was expressed in BL21 (*DE3*) Gold in LB. After reaching O.D ~ 0.6 at

**Table 2.  Primers used during this study.**

| Construct | Forward | Reverse |
|---|---|---|
| pEC-K-3C-His CAL1$_{1-160}$ | 5′-ccaggggcccgactcgatggctaacgcggttg-3′ | 5′-cagaccgccaccgactgcttatttttggcgggctcgc-3′ |
| CAL1$_{1-160-LL-841-979}$ | 5′-ccaggggcccgactcgatggctaacgcggttg-3′<br>5′-gcgagcccgcccaaaaggcagcagcggcggcagcagcggcggtggtgacccggattg-3′ | 5′-cagaccgccaccgactgcttacttatcaccggag-3′<br>5′-caatccgggtcaccaccgccgctgctgccgccgctgctgcctttttggcgggctcgc-3′ |
| Codon Optimised CAL1$_{1-160\ W22A/F29A}$ | 5′-acgaacgttccaaagctgcgtcctctaagatggcgg-3′<br>5′-ggtcctctaagatggcggatgctgcatccctggaagat-3′ | 5′-ccgccatcttagaggacgcagctttggaacgttcgt-3′<br>5′-atcttccagggatgcagcatccgccatcttagaggacc-3′ |
| Codon Optimised CAL1$_{1-160\ W22R/F29R}$ | 5′-ggtatacgaacgttccaaagctaggtcctctaagatg-3′<br>5′-ggtcctctaagatggcggatcgtgcatccctggaagat-3′ | 5′-catcttagaggacctagctttggaacgttcgtatacc-3′<br>5′-atcttccagggatgcacgatccgccatcttagaggacc-3′ |
| Codon Optimised CAL1$_{1-160\ F43R}$ | 5′-gcatggaaattgacgtcgcagaacgcgataacctgttccacgg-3′ | 5′-ccgtggaacaggttatcgcgttctgcgacgtcaatttccatgc-3′ |
| pEC-K-3C-His CAL1$_{1-50}$ | 5′-ccaggggcccgactcgatggctaacgcggttg-3′ | 5′-cagaccgccaccgactgcttattcaccgtggaacagg-3′ |
| 14S CAL1$_{841-979}$ | 5′-tacttccaatccaatgcaatgggtggtgacccggattg-3′ | 5′-ttatccacttccaatgttatta cttatcaccggagttg-3′ |
| Codon Optimised CAL1$_{841-979\ I900R/K907A/Y908A}$ | 5′-acagggcctgggcaaaatcaggggcgaacgttg-3′<br>5′aatcatcggcgaacgttgggcgcgtgcggccctgaaatacacattggtagccgttcctt-3′ | 5′-caacgttcgcccctgattttgcccaggccctgt-3′<br>5′aaggaacggctaccaatgtggtatttcagggccgcacgcgccaacgttcgccgatgatt-3 |
| Non-Codon Optimised CAL1$_{W22A/F29A}$ | 5-gagcgctctaaggccgcgtccagtaaaatggcg-3′<br>5′-ctggtccagtaaaatggcggatgctgccagcctggaa-3′ | 5′-cgccattttactggacgcggccttagagcgctc-3′<br>5′-ttccaggctggcagcatccgccattttactggaccag-3′ |
| Non-Codon Optimised CAL1$_{W22R/F29R}$ | 5′-gagcgctctaaggccaggtccagtaaaatgg-3′<br>5′-ctggtccagtaaaatggcggatcgtgccagcctggaa-3′ | 5′-ccattttactggacctggccttagagcgctc-3′<br>5′-ttccaggctggcacgatccgccattttactggaccag-3′ |
| Non-Codon Optimised CAL1$_{F43R}$ | 5′-gaaatagatgtggccgagcgcgacaacttgttccacgg-3′ | 5′-ccgtggaacaagttgtcgcgctcggccacatctatttc-3′ |
| Non-Codon Optimised CAL1$_{I900R/K907A/Y908A}$ | 5′-cgagcagggactcggaaagattaggggagaacgttggg-3′<br>5′-ggagaacgttgggcccgcgcggccctgaagtaccacatcgg-3′ | 5′-cccaacgttctcccctaatctttccgagtcctgctcg-3′<br>5′ccgatgtggtacttcagggccgcgcgggcccaacgttctcc-3′ |
| CENP-A$_{101-225\ S154Q}$ | 5′-ccaagctgccgttccagcgtctagtgcgcg-3′ | 5′-cgcgcactagacgctggaacggcagcttgg-3′ |
| CENP-A$_{101-225\ M186A}$ | 5′-caggagtcgtgcgaggcgtacttgacgcagcg -3′ | 5′-cgctgcgtcaagtacgcctcgcacgactcctg -3′ |
| CENP-A$_{101-225\ Q190G}$ | 5′-gagatgtacttgacggggcggctcgccgactc-3′ | 5′-gagtcggcgagccgcccccgtcaagtacatctc-3′ |
| CENP-A$_{101-225\ M186A\ Q190G}$ | 5′-caggagtcgtgcgaggcgtacttgacggggcg-3′ | 5′-cgccccgtcaagtacgcctcgcacgactcctg-3′ |
| pEC-K-3C-His CENP-C$_{1264-1411}$ | 5′-ccaggggcccgactcgatgggcccggtagtgttc-3′ | 5′-cagaccgccaccgactgcttaagaacggatacac-3′ |

37°C, cultures were induced at 18°C using 0.3 mM IPTG before being purified under native and denaturing conditions. Under native conditions for all protein, pellets were resuspended in a buffer containing 20 mM Tris–HCl pH 8.0, 100 mM NaCl, 35 mM imidazole and 2 mM βME and supplemented with 1 mM PMSF and cOmplete EDTA-free (Sigma) before lysing by sonication. Clarified lysates were applied onto a HisTrap® HP column. For His-CAL1$_{1-160}$, His-CAL1$_{1-50}$ and His-CAL1$_{1-160-LL-841-979}$, HisTrap® HP columns were then washed with 60 CV of lysis buffer, 20 CV of 20 mM Tris–HCl pH 8.0, 1 M NaCl, 35 mM imidazole, 50 mM KCl, 10 mM MgCl$_2$, 2 mM ATP and 2 mM βME and then 10 CV of lysis buffer. Proteins were then eluted in 20 mM Tris–HCl pH 8.0, 100 mM NaCl, 500 mM imidazole and 2 mM βME. For His-SUMO-CAL1$_{841-979}$ and His-CENP-C$_{1264-1411}$, HisTrap® HP columns were washed with 80 CV of lysis buffer. Protein was eluted using 20 mM Tris–HCl pH 8.0, 100 mM NaCl, 500 mM imidazole and 2 mM βME, and fractions containing protein were dialysed overnight against 20 mM Tris–HCl pH 8.0, 500 mM NaCl and 2 mM DTT for His-CENP-C$_{1264-1411}$ and 20 mM Tris–HCl pH 8.0, 100 mM NaCl and 2 mM DTT for His-SUMO-CAL1$_{841-979}$. His-SUMO-CAL1$_{841-979}$ was applied to a HiTrap® Q HP column and eluted with a gradient of 20 mM Tris–HCl (pH 8.0), 1 M NaCl and 2 mM DTT.

Tags were removed by incubation with 3C or TEV overnight followed by a reverse affinity step to remove His-SUMO. Proteins were purified by SEC using either a Superdex 200 increase 10/300 GL, Superdex 75 10/300 GL or Superdex 75 increase 10/300 GL column (GE Healthcare).

For denatured His-CAL1$_{1-160}$, His-CAL1$_{1-50}$ and His-CAL1$_{1-160-LL-841-979}$, pellets were suspended in 2 ml/g of wet pellet of 20 mM Tris–HCl pH 8.0, 500 mM NaCl, 25 mM imidazole, 7 M urea and 2 mM βME, and incubated for 1 h at 4°C with rotation. DNA was sheared by sonication before clarifying by centrifugation. Lysate was then incubated with 10 ml of HisPur™ Ni-NTA resin (Thermo Fisher Scientific) overnight, before washing with 60 CV of buffer, 20 CV of 20 mM Tris–HCl (pH 8.0), 1 M NaCl, 25 mM imidazole, 7 M urea and 2 mM βME, 10 CV of 500 mM NaCl buffer before eluting with 20 mM Tris–HCl (pH 8.0), 500 mM NaCl, 500 mM imidazole, 7 M guanidine HCl and 2 mM βME.

### Protein refolding

To refold histones with and without CAL1, histones were resuspended in 20 mM Tris–HCl pH 7.5, 7 M guanidine HCl and 2 mM βME and mixed with equimolar amounts of proteins needed. Proteins were then dialysed for 2 h at 4°C against 200 ml of 20 mM Tris–HCl pH 7.5, 7 M guanidine HCl and 2 mM βME; then, 2 l of 10 mM Tris–HCl pH 7.5, 2 M NaCl, 1 mM EDTA and 5 mM βME was slowly added overnight using a peristaltic pump. If needed, refolded protein was further dialysed against a lower salt concentration solvent; if not, complexes were purified by SEC using either a

Superdex 200 increase 10/300 GL or Superdex 75 increase 10/300 GL column (GE Healthcare).

### Crystallisation

Crystallisation trials were performed using a nanolitre Crystal Gryphon robot (Art Robbins) and grown by vapour diffusion methods. For $CAL1_{1-160}$-$CENP$-$A_{101-225}$-H4, 27 mg/ml of complex in a buffer of 20 mM Tris–HCl pH 8.0, 1 M NaCl and 5 mM DTT crystallised after about a year in 0.01 M Cobalt (II) chloride hexahydrate, 0.1 M MES pH 6.5 and 1.8 M ammonium sulphate at 18°C. For $CAL1_{1-160}$-$CENP$-$A_{144-225}$-H4, protein in 20 mM Tris–HCl pH 8.0, 1 M NaCl and 2 mM DTT was concentrated to 17 mg/ml and crystallised in C11, 0.2 M lithium sulphate, 0.1 M Tris pH 8.5 and 30% PEG 4000. An optimisation screen was set up in a 24-well format, using half the original concentrations of protein. Tris–HCl pH 8.5 was kept at 0.1 M, while concentrations of lithium sulphate varied from 0.1 to 0.3 M and PEG 4000 varied from 24 to 34%.

Cleaved $CENP$-$C_{1264-1411}$ was screened at 15 mg/ml in 20 mM Tris–HCl pH 8.0, 500 mM NaCl and 2 mM DTT against several commercial and homemade screens at 18°C. Crystals were obtained in around 13% of all conditions tested. His-$CENP$-$C_{1264-1411}$-$CAL1_{841-979}$ complex was made in 20 mM Tris–HCl pH 8.0, 100 mM NaCl and 2 mM DTT and used with Structure 1 + 2 and JCSG+ (Molecular Dimensions) at 15 mg/ml at 4°C. Crystals were briefly transferred to a cryoprotectant solution (either oil or the mother liquor supplemented with 40% peg 3350) before directly flash cooled in liquid nitrogen and analysed on beamlines i03 and i04-1 at the Diamond Light Source (Didcot, UK).

### Data collection and crystal structure determination

Diffraction data were collected on beamlines i03 ($CAL1_{1-160}$–$CENP$-A/H4 form I, $CENP$-$C_{1264-1411}$; $CENP$-$C_{1264-1411}$-$CAL1_{841-979}$) and i04-1 ($CAL1_{1-160}$–$CENP$-A/H4 form II), at the Diamond Light Source (Didcot, UK). Data were processed using the software pipeline available at Diamond Light Source that relies on XDS, CCP4, CCTBX, AutoPROC and STARANISO (Grosse-Kunstleve *et al*, 2002; Kabsch, 2010; Vonrhein *et al*, 2011; Winn *et al*, 2011; Winter & McAuley, 2011; Tickle *et al*, 2018). X-ray diffraction of form II crystal was highly anisotropic. Processing the same dataset using Xia2 with a CC1/2 cut-off value of 0.3 only included data upto 5.4 Å, while usable data extended upto 4 Å along the c* axis. Hence, STARANISO of AutoPROC was used to analyse the diffraction intensities and to apply an aniosotropic cut-off and correction. STARANISO-defined resolution limits of ellipsoid fitted to resolution cut-off surface are 7.1, 7.1 and 4.1 Å, along *a**, *b** and *c** axes, respectively. $CAL1_{1-160}$–$CENP$-A/H4 (forms I and II), $CENP$-$C_{1264-1411}$ and $CENP$-$C_{1264-1411}$-$CAL1_{841-979}$ structures were determined by molecular replacement with the program PHASER (McCoy *et al*, 2007) using the coordinates of *dm* H3/H4 heterodimer deduced from the structure of *dm* nucleosome core particle, PDB: 2PYO (Clapier *et al*, 2008) and budding yeast Mif2p cupin domain, PDB: 2VPV (Cohen *et al*, 2008) and *dm* $CENP$-$C_{1264-1411}$ determined here, respectively. Structures were refined using the PHENIX suite of programs (Adams *et al*, 2010). $CAL1_{1-160}$–$CENP$-A/H4 form II was refined using PHENIX-Rosetta (DiMaio *et al*, 2013). Model building and structural superpositions were done using COOT (Emsley & Cowtan, 2004).

Figures were prepared using PyMOL (http://www.pymol.org). Data collection, phasing and refinement statistics are shown in Table 1.

### Ni-NTA interaction trials

Ni-NTA pull-down assays were performed using His-$CAL1_{1-160\ WT}$ and mutants mixed with 1.3 times molar excess of $CENP$-$A_{101-225}$-H4 and made up to 200 µl with 20 mM Tris–HCl pH 8.0, 2 M NaCl, 10% glycerol, 0.5% NP40, 35 mM imidazole and 2 mM βME. 190 µl was incubated with 120 µl of HisPur™ Ni-NTA resin slurry that had been washed with ddH$_2$O and buffer for 30 min at 4°C. Beads were then washed four times with 1 ml of buffer, then twice with 1 ml of 20 mM Tris–HCl pH 8.0, 500 mM NaCl, 35 mM imidazole and 2 mM βME and eluted by boiling in SDS–PAGE loading dye before being separated on a Bolt™ 4–12% Bis-Tris Plus gel (Invitrogen) run at 180 V for 1 h in MES buffer. Gels were then stained with Coomassie Blue, and scanned gel images were analysed and quantified with ImageJ.

### SEC-MALS

Size-exclusion chromatography (ÄKTA-Micro™, GE Healthcare) coupled to UV, static light scattering and refractive index detection (Viscotek SEC-MALS 20 and Viscotek RI Detector VE3580; Malvern Instruments) was used to determine the molecular mass of proteins and protein complexes in solution. Injections of 100 µl of 1–5 mg/ml material were used.

For His-$CAL1_{1-160}$-$CENP$-$A_{101-225}$-H4 ($\partial A_{280\ nm}/\partial c = 0.67$ AU.ml/mg), His-$CAL1_{1-160}$-$CENP$-$A_{144-225}$-H4 ($\partial A_{280\ nm}/\partial c = 0.75$ AU.ml/mg) and His-$CAL1_{1-50}$-$CENP$-$A_{101-225}$-H4 ($\partial A_{280\ nm}/\partial c = 0.55$ AU.ml/mg) were run on a Superdex 200 increase 10/300 GL size-exclusion column pre-equilibrated in 50 mM HEPES pH 8.0, 2 M NaCl and 1 mM TCEP at 22°C with a flow rate of 0.5 ml/min. His-$CENP$-$C_{1264-1411\ L1357E/M1407E}$ ($\partial A_{280\ nm}/\partial c = 0.75$ AU.ml/mg) was run on a Superdex 200 increase 10/300 GL size-exclusion column pre-equilibrated in 50 mM HEPES pH 8.0, 300 mM NaCl and 1 mM TCEP at 22°C with a flow rate of 0.5 ml/min. $CENP$-$C_{1264-1411}$ ($\partial A_{280\ nm}/\partial c = 0.84$ AU.ml/mg) was run at 4°C on a Superdex 75 increase 10/300 GL size-exclusion column pre-equilibrated in 50 mM HEPES pH 8.0, 100 mM NaCl and 1 mM TCEP.

Light scattering, refractive index (RI) and $A_{280\ nm}$ were analysed by a homo-polymer model (OmniSEC software, v5.02; Malvern Instruments) using the parameters stated for each protein, $\partial n/\partial c = 0.185$ ml/g and buffer RI value of 1.335. The mean standard error in the mass accuracy determined for a range of protein–protein complexes spanning the mass range of 6–600 kDa is $\pm$ 1.9%.

### Cross-linking mass spectrometry

Cross-linking was performed on gel-filtered complexes dialysed into PBS. 30 µg EDC (Thermo Fisher Scientific) and 66 µg sulpho-NHS (Thermo Fisher Scientific) were used to cross-link 10 µg of protein for 1.5 h at RT. 30 µg of BS$^3$ was used to cross-link 10 µg of protein for 2 h at RT. The reactions were quenched with final concentration 100 mM Tris–HCl or 5 mM ammonium bicarbonate, respectively, before separation on Bolt™ 4–12% Bis-Tris Plus gels (Invitrogen). Following previously established protocol (Maiolica *et al*, 2007), the bands were excised and proteins were digested with 13 ng/µl

trypsin (Pierce) overnight at 37°C after being reduced and alkylated. The digested peptides were loaded onto C18-Stage-tips (Rappsilber *et al*, 2007) for LC-MS/MS analysis (see Appendix Supplementary Methods).

## Cell culture and transfections

Schneider S2 cells containing the LacO array (L2-4_LacO_LexA_-Clone11) were generated as described in Mendiburo *et al* (2011). Schneider S2 cells were grown at 25°C in Schneider's Drosophila medium (Gibco) supplemented with 10% foetal calf serum (Sigma). Cells were seeded in 24-well plates, a day prior to transfection at a density of $5 \times 10^5$ cells per well. Cells were transfected using X-tremeGENE DNA Transfection Reagent (Roche) according to the manufacturer's instructions, using 200 ng of plasmid DNA. Transfected cells were analysed by immunofluorescence 72 h post-transfection.

U2OS cells containing 200 copies of an array of 256 tandem repeats of the 17 bp LacO sequence on chromosome 1 (gift from B.E. Black, University of Pennsylvania, Philadelphia; Janicki *et al*, 2004) were grown in DMEM supplemented with 10% FBS and 1% penicillin–streptomycin at 37°C in a 5% $CO_2$ incubator. Cells were seeded in 10 cm dishes, a day prior to transfection at a density of $2.5 \times 10^6$ cells per well. Transfections were performed with Lipofectamine 3000 (Life Technologies) according to the manufacturer's instructions, using 15 µg of plasmid DNA and Opti-MEM I reduced serum medium (Life Technologies). Next day, cells were washed once with 1×DPBS, trypsinised, counted and re-plated on poly-lysine-coated coverslips in 6-well plates at a density of $10^6$ cells per well. Downstream experiments were performed 3 days post-transfection.

## Immunofluorescence

Cells were washed once in PBS and then fixed with 3.7% formaldehyde in 0.1% Triton X-100 in 1× PBS (PBST) for 8 min at RT. Following fixation, the slides were washed once in PBST and then blocked in Image-iT® FX signal enhancer in a humidified chamber at RT for at least 30 min. All antibodies were incubated in a 1:1 mix of PBST and 10% normal goat serum (Life Technologies) overnight at 4°C in a humidified chamber and were used in 1:100 dilution unless otherwise stated: myc (Abcam-ab9106), V5 (Invitrogen-R96025) and HA (clone 3F10; E. Kremmer, 1:20). Secondary antibodies coupled to Alexa Fluor 555 and 647 (Invitrogen) were used at 1:100 dilutions. Counterstaining of DNA was performed with DAPI (5 µg/ml), and coverslips were mounted on the slides with 30 µl of SlowFade® Gold antifade reagent.

## Microscopy and image analysis

All IF images were taken as 50 *z*-stacks of 0.2 µm increments, using a 100× oil immersion objective on a DeltaVision RT Elite Microscope and a CoolSNAP HQ Monochrome camera. All images were deconvolved using the aggressive deconvolution mode on a SoftWorx Explorer Suite (Applied Precision) and are shown as quick projections of maximum intensity. For U2OS cells, the mean fluorescence intensity of the protein of interest was measured at the LacO spot, and then the mean fluorescence intensity in the nucleus (background) was subtracted from this value. For S2 cells, the mean fluorescence intensity of the protein of interest was measured at the LacO spot, and then the mean fluorescence intensity of three spots around the LacO spot was subtracted from this value. 25–50 cells were analysed per biological replicate, and a minimum of three independent biological replicates were quantified per experiment.

## Analytical ultracentrifugation

Sedimentation velocity and SE experiments were performed using a Beckman Coulter XL-I analytical ultracentrifuge equipped with an An-50 Ti eight-hole rotor. Depending on their concentration, samples were loaded into 12 (low concentration) or 3 mm (high concentration) pathlength charcoal-filled epon double-sector centre-pieces, sandwiched between two sapphire windows. For SV, samples were equilibrated at 4°C in vacuum for 6 h before running at 49 k rpm. For SE, data were recorded at 26 k rpm. The laser delay, brightness and contrast were pre-adjusted at 3 k rpm to acquire the best quality interference fringes. Data were collected using Rayleigh interference and absorbance optics recording radial intensity or absorbance at 280 nm. For SV, data were recorded between radial positions of 5.65 and 7.25 cm, with a radial resolution of 0.005 cm and a time interval of 7 min, and analysed with the program SEDFIT (Schuck, 2000) using a continuous c(s) model. For SE, data were recorded between radial positions of 6.00 and 7.25 cm, with a radial resolution of 0.001 cm and a time interval of 3 h (until successive scans overlaid satisfactorily), and analysed with the program SEDPHAT (Vistica *et al*, 2004) using species analysis. The partial specific volume, buffer density and viscosity were calculated using SEDNTERP (Hayes *et al*, 2012). Sedimentation coefficients were computed from atomic coordinate models using SOMO (Brookes & Rocco, 2018).

## Statistics

Data are representative of at least three independent experiments, unless otherwise stated. Statistics were performed using GraphPad Prism, version 7.0e (GraphPad Software, Inc), using an unpaired two-tailed *t*-test or Mann–Whitney test. For each statistical test, *P*-value of < 0.05 was considered significant.

# Data availability

The structural coordinates and structure factors reported in this paper have been deposited in the PDB (http://www.rcsb.org/) with the following accession numbers: 6XWT—CAL1$_{1-160}$–CENP-A/H4 form I, 6XWS—CAL1$_{1-160}$–CENP-A/H4 form II, 6XWU—CENP-C$_{1264-1411}$ and 6XWV—CENP-C$_{1264-1411}$-CAL1$_{841-979}$. Protein cross-linking/mass spectrometry data have been deposited in PRIDE (www.ebi.ac.uk/pride/) with accession number PXD017238.

**Expanded View** for this article is available online.

## Acknowledgements

We thank the staff of the Edinburgh Protein Production Facility, especially Martin Wear, for their help with SEC-MALS and SPR. Thanks to Atlanta Cook for critical reading of the manuscript. The Wellcome Trust generously

supported this work through a Wellcome Trust Career Development Grant (095822) and a Senior Research Fellowship (202811) to AAJ, a Senior Research Fellowship (084229) to JR, a Senior Research Fellowship (103897) to PH, a Centre Core Grant (092076 and 203149) and an instrument grant (091020) to the Wellcome Trust Centre for Cell Biology, a Multi-User Equipment grant 101527/Z/13/Z to the EPPF and a Wellcome-UoE ISSF award towards the procurement of SEC-MALS equipment for the EPPF. PH was further supported by a European Research Council Starting-Consolidator Grant (311674– BioSynCEN).

## Author contributions

AAJ and PH conceived the project. BM-P, VL, PH and AAJ designed experiments. BM-P,VL, OB, MAA and JZ performed experiments. JR provided expertise and feedback. BM-P and AAJ wrote the manuscript with input from all authors.

## Conflict of interest

The authors declare that they have no conflict of interest.

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
