## [Review Process File · The EMBO Journal]

Structural Basis for Centromere Maintenance by *Drosophila* CENP-A Chaperone CAL1

Bethan Medina-Pritchard, Vasiliki Lazou, Juan Zou, Olwyn Byron, Maria Abad, Juri Rappsilber, Patrick Heun, and A. Arockia Jeyaprakash.

Review timeline:	Submission date:	15 th August 2019
	Editorial Decision:	2 nd October 2019
	Revision received:	25 th January 2020
	Accepted:	11 th February 2020

Editor: Hartmut Vodermaier

Transaction Report:

1st Editorial Decision

2nd October 2019

Thank you again for submitting your manuscript on centromeric histone chaperone CAL1 structures for our editorial consideration, and apologies once more for its delayed evaluation. We have now finally received a complete set of reports from three expert referees, copied below for your information. Given the overall positive nature of their feedback, we would be interested in considering a revised manuscript further for eventual EMBO Journal publication. Acceptance would however depend on satisfactorily addressing a number of major points raised by the referees, particularly the structural concerns raised by referee 1, and referee 2's criticisms regarding 'in vivo' validation, which should be extended to testing key mutants in a more physiologically relevant system (e.g. RNAi/rescue experiments in S2 cells instead of LacI tethering in heterologous systems).

REFeree REPORTS

Referee #1:

Summary: The authors crystallized the CAL1 N-terminal fragments with CID/H4 (N-terminal tail deletion of CID), as well as the CAL1-C-terminal fragment with the CENP-C C-terminal fragment. Results support the lack of requirement for a Mis18 complex in *Drosophila*, and suggest some new aspects of CENP-A loading that may be significant in humans, i.e. cross talk in recruitment between CENP-C and CAL1/HJURP. Mutational analyses and biophysical validation of complex stoichiometry agree with crystallographic results, and expected recruitment phenotypes in live cells.

Positive:

- Resolution on 'Form 1' crystal (3.47Å) 'good enough' for interface side chains, shows only residues 18-46 (or 14-48? unclear) of CAL1
- Conformation of CENP-A/H4 agrees pretty well with H3/H4 (differences likely due to CAL1 binding) and good superimposition with HJURP/CENP-A/H4 and Scm3/CENP-A/H4 structures.

- Mutations of key residues observed in crystal structure result in reduced apparent affinity in vitro (pull downs, SEC-MALS) and localization in live cells (fluorescent tagged fusion, recruiting to Lac operon)
- CX-MS supports some of the crystal contacts, and helped make sense of lower resolution electron density in crystal form 2, showing more of the CAL1 N-terminus wrapping around the rest of the CENPA/H4
- Form II structure implies occlusion of CENP-A/H4 tetramerization, H2AB binding, and DNA binding interfaces when bound to CAL1, in agreement with literature observations in *Drosophila* and humans
- CAL1 recognition is structurally but not sequence conserved with HJURP/Scm3, which was implied by recruitment assays using CENP-A and H3 hybridizations
- CENP-C crystal structure confirms predicted formation of a cupin domain which dimerizes; Dimerization requirement for CAL1 fragment binding implies stoichiometry that may be crucial for CENP-A deposition, may also impact recruitment
- CENP-C and CENP-C/CAL1 crystals have good Rwork/Rfree and B factors
- Mutations based on CENP-C/CAL1 structure interface at least weakened binding, stoichiometry supported by AUC

Criticism:

- Call-histone structures are of limited resolution, and the statistics especially for Form II are a bit problematic. B-factors are very high. Based on the I/σ values in the highest resolution shell, it looks like a very conservative resolution cut-off was applied (especially for the P 63 2 2 dataset, with I/σ of 2.4). A better measure of the information content of a diffraction dataset is the CC1/2, and it is often the case that you get good CC1/2 even at resolutions where I/σ is lower than 2, so the current best practice is to report a CC1/2 value and to base your resolution cut-off on this criterion, not on I/σ (or Rmerge or Rpim). These two papers from 2012 and 2013 explain CC1/2: <https://doi.org/10.1126/science.1218231> and <https://doi.org/10.1107/S0907444913001121>. So, you might want to take a look at these papers, report a CC1/2 (at least XDS will report it, probably other data processing programs as well) and base your resolution cut-off on the resulting value. You might get a little bit more out of their datasets, which may improve your models.
- What's up with the low data completeness for form 2? Is this a mistake? I am not sure how valid the structure is; also in light of the high B-factor.
- This statement should not be made: 'While side chain electron densities are well defined for CAL1 residues 7 to 47, only the main chain could be modelled for rest of the CAL1' for resolution of 4.3 Å. I didn't see any well defined side chains.
- When discussing the structure of the histone complex, which crystal form do they refer to? and how similar are the two crystal forms?
- Nothing super novel beyond confirmation of literature idea that divergent CAL1 bridges HJURP and Mis18 activities on its own. Recognition by CAL1 is a little different, but CENPA/H4 conformation is almost exactly the same in humans and *D.me*
- CX-MS is missing most of the key contacts (from form I) in the crystal structure, instead adjacent regions crosslinked. This should be explained. Maybe form 1 and form 2 aren't related.
- Some conclusions drawn from ~4.5Å structure without side chain densities, but CX-MS backs those up
- CAL1/CENP-A/H4 structures have high B-factor, but they took a year to crystallize so I doubt much optimization is realistic, and density looks good around the interaction interface.

Minor quibbles

- Needs some more proofreading, in several spots amino acid #s contradict text/text or text/figure, a couple are just typos in the truncation numbers but at least 1 crucial interaction interface residue is numbered wrong when referring to figure 6A.
- Overall the discussion of the structure with histones is a bit long.
- There are NO page numbers.
- Molecular weights from SEC-MALS and AUC are reported without any confidence statistics
- Summary: this statement is inaccurate 'using crystal structures, we show how *Drosophila* CAL1, an evolutionarily distinct CENP-A chaperone, targets CENP-A to the centromere receptor CENP-C without the requirement of the Mis18 complex'. The crystal structure shows nothing about targeting, that is the interpretation.
- Maybe a bit over stated 'This observation is in striking agreement with the subunit stoichiometry of the human pre-nucleosomal CENP-A/H4 in complex with HJURP' - what's so striking about that?
- this statement in the discussion is strange, in light of the similarities with HJURP: The overall

mode of CENP-A/H4 recognition by CAL1 appears to be novel'; especially since they state in the text: 'Structural superposition of CAL1-CENP-A/H4 onto its respective human and *Kluyveromyces lactis* structures, HJURP-CENP-A/H4 (PDB: 3R45) (Hu et al., 2011) and Scm3-CENP-A/H4 (PDB: 2YFV) (Cho and Harrison, 2011) showed that CAL1 employs a broadly similar mode of CENP-A recognition with a few striking differences (Fig. 3A). This needs to be phrased more carefully.

Referee #2:

The centromere is specified by sequence-independent epigenetic mechanisms, and histone H3 variant CENP-A plays an epigenetic mark for centromere specification. Therefore, it is a central subject in the centromere research field how CENP-A is deposited into centromeres. In human, CENP-A deposition is mediated by the CENP-A-specific chaperone HJURP and the chromatin licensing factor Mis18 complex. However, in *Drosophila*, there are no HJURP and Mis18 complex, although CENP-A is conserved. Instead, *Drosophila* specific factor Cal1 seems to mediate CENP-A deposition in *Drosophila* cells. However, it remains largely unknown how Cal1 mediates CENP-A deposition in *Drosophila* cells. In this study, authors analyzed crystal structures of the Cal1-CENP-A-H4 complex and demonstrated that the Cal1 N-terminal region binds to mainly CENP-A $\alpha 2$ helix, as HJURP does. However, the binding mode of Cal1 to CENP-A is slightly different from that of HJURP. Important residues for Cal1-CENP-A-H4 interaction observed in *in vitro* studies were confirmed by *in vivo* analyses.

In addition to the Cal1-CENP-A-H4 complex, authors also solved the structure of the Cal1-CENP-C complex in which the Cal1 C-terminal region binds to CENP-C dimer domain (Cupin domain). Combining two fine structures (Cal1-CENP-A-H4 and Cal1-CENP-C), the authors propose that Cal1 plays a critical function for CENP-A deposition in *Drosophila* cells.

Overall, the structural studies are solid. Although previous studies predicted that Cal1 has similar activity of HJURP, this study clearly demonstrate that Cal1 binds to CENP-A-H4 using a similar way as HJURP at atomic resolution. It is an important step in the centromere research field. The structure of Cal1-CENP-C structure also gives us important information. We highly appreciate the challenge to the authors. However, I have several technical concerns, and the authors should address them prior to publication. Especially, I do not think that experiments with expression of *Drosophila* proteins in U2OS cells are *in vivo* studies. If they want to examine protein interactions *in vivo*, they should use *Drosophila* cells. My specific concerns are following:

1. As mentioned above, the experiments in Fig2D and Fig7D are not real *in vivo* studies. As other centromere (and/or regulatory) factors might not interact with *Drosophila* CENP-A, Cal1, and CENP-C in U2OS cells, these experiments are essentially similar to *in vitro* studies. Experiments themselves are slightly messy. If the authors want to demonstrate interaction *in vivo*, they should use *Drosophila* culture cells. Alternatively, they can show important residues in flies.
2. Concerning section "Cal1 uses conserved and adaptive interaction...", authors mentioned "CENP-A at position 86". Is this true? It might be 186. In addition, there are no alignments of human, yeast CENP-A and *Dm* CENP-A. Author should show these alignments, if they discuss about conservation of human yeast and *Drosophila* CENP-A.
3. Authors demonstrate that the $\alpha 1$ helix of Cal1 is swinging away compared with HJURP and $\alpha 5$ and $\alpha 6$ help binding of Cal1 to CENP-A-H4. This is a really interesting observation. Can they compare affinity of Cal1 to CENP-A-H4 in the presence or absence of $\alpha 5$ and $\alpha 6$? I think that Cal1 (1-50) might lack $\alpha 5$ and $\alpha 6$. How about its affinity compared with Cal1 (1-160)?
4. Crystal structures of *S. pombe* and *Drosophila* Cupin domain were reported recently (Chik JK et al., JBC, 2019, PMID: 31366733). Authors should cite this reference and discuss similarities and differences. In addition, the *S. pombe* Cupin domain binds to Moa1, and authors should mention whether similar binding region is used for Cal1 binding for the CENP-C Cupin domain.
5. While I understand expression of full length Cal1 is really hard, I am very curious whether Cal1 binds to CENP-A and CENP-C simultaneously. Although it might be hard to test this with *in vitro* experiments, if they can express tagged Cal1 in *Drosophila* cells and perform IP experiments, they might address this point. If they added such a experiments, quality of the paper would increase.

Referee #3:

This manuscript provides novel structural insights into the molecular interactions between the three key chromatin components of the *Drosophila* centromere; CENP-A, its chaperone CAL1 and the CENP-A and CAL1 binding protein CENP-C. These three proteins have been proposed to engage in self-enforcing tripartite positive feedback loop that may be sufficient, or is central to, centromere maintenance in fly cells. Some of the key interactions have been mapped earlier and proposed in work by the same team in a recent report that is published as a preprint.

What they now add is further crystallographic characterization of the CAL1 domain that binds to CENP-A/H4 dimers. While the crystals obtained offered a modest resolution, they allowed to trace the backbone of CAL1 across most of the interaction surface on the CENP-A dimer. This provides insight into the conserved nature of CENP-A binding to its chaperone. While structures have been solved for both human and yeast proteins, this addition is relevant as CAL1 is evolutionally diverged, yet as is shown here, interacts with CENP-A pre-nucleosomes in a quite similar manner. The authors further show sedimentation data to determine protein stoichiometries and show a single CAL1 protein binds a single CENP-A/H4 heterodimer. Mutational analysis identified several key residues, although it is clear from this that the interaction of CAL1 with the histone dimer is multivalent, and many sites need to be altered to impact binding.

Further, the structure of the CENP-C cupin domain was solved that was previously shown to interact with CAL1. Of interest here is that the binding of CAL1 requires the CENP-C cupin domain to dimerize and mutations in the dimerization domain of CENP-C abrogate CAL1 binding, both in vitro and on LacO arrays in cells. The work here offers the a structural basis for this.

While I am not a crystallographer, and cannot comment on the quality of the molecular models derived from the densities that were obtained, I believe that the work is both novel and very insightful offering new insight into the mechanisms of how CENP-A chromatin can be self-templating and epigenetically maintained.

The work is a comprehensive body of work combining X-ray crystallography, in vitro binding and stoichiometry analysis, in vivo binding assays and mutational analysis that make a convincing case for their claims.

I support publication overall but have several comments that need to be addressed.

Figure 2A, F43 is discussed in the text but not indicated in the figure. It would be helpful to have this shown.

Figure 2C: Is there a particular reason why the W22R and F29A/R single mutants are not tested?

Figure 3. The CAL1 CENP-/H4 structure is compared with corresponding structures of HJURP and Scm3. It would help to indicate what fragments of these proteins were resolved in these structures. Is it possible that some of the difference observed with CAL1 are due to the fact that different sized fragments are compared? This should be acknowledged.

Figure 3C. The amount of CENP-A/H4 in the input is quite variable which makes interpretation of this experiment difficult. A repeat of this with more normalized starting material should be performed and quantification of the binding should be included.

The models in figure 4 are very busy and it is hard to see the sterical clashes that CAL1 would cause. I suggest including here images of just the CENP-A/H4 heterotetramer (no H2A and H2B and no DNA) to visualize better how CAL1 prevents CENP-A/H4 tetramerization.

Figure 5 would be more legible if a linear diagram of CENP-C is shown mapping the domain that is solved and showing the position of the Cupin domain (similar to the schematics in Figure 1A).

Figure 7. The CENP-C residues M1407 and L1357 are important for CENP-C dimerization and in this figure they are shown to be also critical for CAL1 binding. The authors interpret this as dimerization of CENP-C being required for CAL1 binding. Alternatively, the CENP-C mutations directly affect CAL1 interactions. How are these residues positioned in the structure relative to the CAL1 binding site? Is it possible that the M1407E and L1357E directly affect CAL1 interactions? If not, this can be highlighted in the figure.

Halfway through the discussed it is stated that CENP-C is present in all organisms with monocentric

chromosomes. However, kinetoplasts have monocentric, regional centromeres but lack CENP-C.

While the single CAL1 binding to a CENP-C dimer is very interesting as it may provide specificity for the centromere, the idea that CAL1 can bring in CENP-C in turn is more vague. The authors suggest that CAL1 dimers or oligomers may bring in CENP-C. However in their previous work (Roure 2019, bioRxiv) they show that CAL1 can bring CENP-C to chromatin but only in the presence of CENP-A. This would suggest that the role for CAL1 in CENP-C recruitment is indirect via CENP-A nucleosome. It is therefore not clear whether the CAL1 CENP-C interaction is sufficient to recruit CENP-C to chromatin. While the authors may choose to discuss this as a hypothetical scenario, I'm very much opposed to the term CENP-C "loading" or CAL1 being a CENP-C "loader". Loading implies more than just binding. It indicates a multistep process to transition from one state to another. There is no evidence that CAL1 is a chaperone for CENP-C that needs to be handed off to another binding partner or DNA. CENP-C simply binds to CENP-A and it binds to CAL1. I urge to authors to stick to the word "binding" not loading.

Referee #1:

Summary: The authors crystallized the CAL1 N-terminal fragments with CID/H4 (N-terminal tail deletion of CID), as well as the CAL1-C-terminal fragment with the CENP-C C-terminal fragment. Results support the lack of requirement for a Mis18 complex in *Drosophila*, and suggest some new aspects of CENP-A loading that may be significant in humans, i.e. cross talk in recruitment between CENP-C and CAL1/HJURP. Mutational analyses and biophysical validation of complex stoichiometry agree with crystallographic results, and expected recruitment phenotypes in live cells.

Positive:

- Resolution on 'Form 1' crystal (3.47Å) 'good enough' for interface side chains, shows only residues 18-46 (or 14-48? unclear) of CAL1
- Conformation of CENP-A/H4 agrees pretty well with H3/H4 (differences likely due to CAL1 binding) and good superimposition with HJURP/CENP-A/H4 and Scm3/CENP-A/H4 structures.
- Mutations of key residues observed in crystal structure result in reduced apparent affinity in vitro (pull downs, SEC-MALS) and localization in live cells (fluorescent tagged fusion, recruiting to Lac operon)
- CX-MS supports some of the crystal contacts, and helped make sense of lower resolution electron density in crystal form 2, showing more of the CAL1 N-terminus wrapping around the rest of the CENP-A/H4
- Form II structure implies occlusion of CENP-A/H4 tetramerization, H2AB binding, and DNA binding interfaces when bound to CAL1, in agreement with literature observations in *Drosophila* and humans
- CAL1 recognition is structurally but not sequence conserved with HJURP/Scm3, which was implied by recruitment assays using CENP-A and H3 hybridizations
- CENP-C crystal structure confirms predicted formation of a cupin domain which dimerizes; Dimerization requirement for CAL1 fragment binding implies stoichiometry that may be crucial for CENP-A deposition, may also impact recruitment
- CENP-C and CENP-C/CAL1 crystals have good R_{work}/R_{free} and B factors
- Mutations based on CENP-C/CAL1 structure interface at least weakened binding, stoichiometry supported by AUC

We thank this reviewer for the overall positive evaluation of our manuscript and for the constructive suggestions.

Criticism:

R1-1. *Cal1-histone structures are of limited resolution, and the statistics especially for Form II are a bit problematic. B-factors are very high. Based on the $1/\sigma$ values in the highest resolution shell, it looks like a very conservative resolution cut-off was applied (especially for the P 63 2 2 dataset, with $1/\sigma$ of 2.4). A better measure of the information content of a diffraction dataset is the CC1/2, and it is often the case that you get good CC1/2 even at resolutions where $1/\sigma$ is lower than 2, so the current best practice is to report a CC1/2 value and to base your resolution cut-off on this criterion, not on $1/\sigma$ (or R_{merge} or R_{pim}). These two papers from 2012 and 2013 explain CC1/2: <https://doi.org/10.1126/science.1218231> and <https://doi.org/10.1107/S0907444913001121>. So, you might want to take a look at these papers, report a CC1/2 (at least XDS will report it, probably other data processing programs as well) and base your resolution cut-off on the resulting value. You might get a little bit more out of their datasets, which may improve your models.*

We thank the reviewer for raising this concern and for the suggestion. We agree that the Pearson correlation coefficient CC1/2 is a robust and statistically informed value that can better define the high-resolution cut-off. However, we could not apply this criterion for the form II dataset for the following

reasons: X-rays diffraction of form II crystal was highly anisotropic. Hence the dataset was processed using AutoPROC (Vonrhein et al., Acta Cryst D 2011), which uses XDS and the scaling programs AIMLESS and POINTLESS for data processing and scaling and STARANISO (Tickle et al., 2019, STARANISO, Cambridge, United Kingdom: Global phasing limited) to analyse the anisotropic diffraction. STARANISO, analyses the merged intensities for anisotropic diffraction, applies an anisotropic cut-off and correction and performs Bayesian estimation of structure amplitudes. STARANISO uses locally-averaged value of $I/\sigma(I)$, rather than its spherical average, to define an anisotropic diffraction cut-off (i.e a cut-off varying with direction in reciprocal space), as a consequence, CC1/2 could not be used as a suitable metric. Processing this data set with Xia2 using conventional spherical resolution cut-off with a CC1/2 of 0.3 includes data only up to 5.4 Å, while diffraction along the c^* -axis extends to 4 Å. We have now explicitly mentioned this in the 'Data Collection and Structure Determination' section of the Materials and Methods (Page 24).

R1-2. What's up with the low data completeness for form 2? Is this a mistake? I am not sure how valid the structure is; also in light of the high B-factor.

We thank the reviewer for bringing this up and we realise that the way the data completeness is presented is misleading. Considering the anisotropic diffraction of form II and for the reasons described above, it would be more appropriate to show the ellipsoidal completeness for this dataset. Table 1 of the revised manuscript now shows the ellipsoidal completeness: Overall 89% and high resolution 81% with a footnote 'ellipsoidal cut-off was determined by STARANISO'. We agree that the average B-factor of the refined structure corresponding to form II is quite high. We have now re-refined this structure with just one refinable isotropic ADP per residue in combination with varying TLS groups. The average B for the latest model is 193 Å². As shown in the Figure R1, prepared for the purpose of this rebuttal, the average B-factor of the latest model (195 Å²) is similar to the mean average B-factor of 131 structures determined at a similar resolution (170.18 Å²) as analysed by Phenix. We believe that the overall model quality agrees with what is expected from a very low-resolution data set (4.4 Å) with a high anisotropic diffraction.

Figure R1

R1-3. *This statement should not be made: ' While side chain electron densities are well defined for CAL1 residues 7 to 47, only the main chain could be modelled for rest of the CAL1' for resolution of 4.3 Å. I didn't see any well-defined side chains.*

We thank the reviewer for this comment and we have now removed this sentence.

R1-4. *When discussing the structure of the histone complex, which crystal form do they refer to? and how similar are the two crystal forms?*

We apologise for the lack of clarity. The CAL1/CENP-A/H4 structures of form I and II superpose with an RMSD of 1.2 Å suggesting that the overall structure of form I and II are essentially the same. However, as the resolution of CAL1/CENP-A/H4 form I is better than of form II, we have now used CENP-A/H4 coordinates of form I for structural comparisons with *Drosophila* H3/H4 and human CENP-A/H4. The structure of *Drosophila* CENP-A/H4 superposes well with both *Drosophila* H3/H4 and human CENP-A/H4 with an RMSD of 1 Å. We have included this information on page 7 & 8.

R1-5. *Nothing super novel beyond confirmation of literature idea that divergent CAL1 bridges HJURP and Mis18 activities on its own. Recognition by CAL1 is a little different, but CENPA/H4 conformation is almost exactly the same in humans and D.me*

While previous experimental evidences suggested CAL1 as a functional equivalent of HJURP that combines the role of human Mis18 in flies, several key questions remained unanswered: 1. How does CAL1 recognize *Drosophila* CENP-A/H4 (structural determinants) and is CAL1 a structural homologue of human HJURP or yeast Scm3? 2. Which regions of CAL1 and CENP-C mediate their interaction and what are the structural determinants mediating CAL1-CENP-C interactions? Our work, while confirming that CAL1 indeed bridges the functions of HJURP and Mis18, also demonstrates: (1) how CAL1, though evolutionarily distinct at the amino acid sequence level employs a conserved (involving the first 50aa) and adaptive (involving 51-160) structural interactions to recognise CENP-A/H4. Particularly, the adaptive interaction involving the CAL1 region downstream of CAL1₁₋₅₀ that wraps around CENP-A/H4 (reported here) appears crucial as CAL1₁₋₅₀ (the Scm3 domain) alone is not sufficient to recruit CENP-A at centromeres and requires downstream residues (Chen et al., 2014 and Roure et al., 2019), (2) how CENP-C cupin domain forms a CAL1 binding surface that is stabilised through cupin dimerization. This is the first structural study providing insights into how an evolutionarily conserved CENP-A receptor CENP-C associates with a CENP-A loading machinery in any organism. The structural characterisations reported here are particularly interesting and insightful considering the minimalistic nature of the number of players involved in CENP-A loading compared to other organisms (such as fission yeast and humans). Hence, we believe these are important contributions to the field of centromere biology.

R1-6. *CX-MS is missing most of the key contacts (from form I) in the crystal structure, instead adjacent regions crosslinked. This should be explained. Maybe form 1 and form 2 aren't related.*

We thank the reviewer for this comment. We have used EDC, a chemical cross-linker that covalently links the carboxylate groups of Asp/Glu (less preferentially hydroxyl group of Ser/Thr) with the primary amine of Lys residues. EDC was our primary choice since it is a zero-length cross-linker that can help identify residues forming the direct binding interface. Unfortunately, the CAL1 N-terminal region (CAL1 1-50) that directly interacts with CENP-A/H4 (in form I) has only a few EDC cross-linkable residues. We think this, together with the availability of tryptic sites in these regions, are the possible reasons for not observing cross-links in these contact sites. However, this approach has revealed cross-links between CAL1 α 1, particularly Ser 19 and Lys 20 and with residues from downstream helical segments (aa 130-160), Glu 139 and Glu 155 and strengthens our conclusion (from form II) that CAL1

1-160 wraps around CENP-A/H4 in a way that brings N and C-terminal regions of CAL1₁₋₁₆₀ in close proximity. By repeating the cross-linking experiment with EDC and by performing new cross-linking experiments with BS₃ (a Lys-Lys cross-linker with a longer spacer) we have confirmed the reliability and reproducibility of these cross-links. We have now included these data in new figure EV3.

A

B

Figure EV3

We believe that crystal forms I and II are closely related, and crystal form I only differs from II in the size of the CAL1 fragment that it associates with, which likely is a result of proteolytic degradation of CAL1₁₋₁₆₀ during the long time the sample took to crystallize. The notion that the interaction between CAL1 residues 14-47 and CENP-A/H4 seen in form I is common to both crystal forms is strengthened by our *in vitro* and *in vivo* validation of structure-guided mutants. CAL1 mutants W22/F29 perturb CENP-A/H4 interaction both in the context of 1-160 and full length CAL1.

R1-7. *Some conclusions drawn from ~4.5Å structure without side chain densities, but CX-MS backs those up*

The crystal form II, although diffracted only to a very low resolution, provides the following structural insights: (1) CAL1 residues 8-30, which forms a single α helical segment ($\alpha 1$), interacts with CENP-A $\alpha 2$ throughout its length very much like HJURP and Scm3. However, the N-terminal half of CAL1 $\alpha 1$ protrudes away from CENP-A $\alpha 2$ and (2) CAL1 α helical segments down stream of $\alpha 1$, critical for CENP-A recruitment in cells, wraps around CENP-A/H4 heterodimer. As these observations concern only gross structural features, we think that this low-resolution structural data, backed up by the complementary Cross-linking MS data, is making useful contributions to fully understand how CENP-A is recognized by CAL1.

R1-8. *CAL1/CENP-A/H4 structures have high B-factor, but they took a year to crystallize so I doubt much optimization is realistic, and density looks good around the interaction interface.*

We agree with the reviewer that the CAL1/CENP-A/H4 structures have high B-factor. Unfortunately, our extensive crystallisation efforts with varying protein concentrations, buffer composition, length of the individual protein fragments and crystallisation temperature were unfortunately not successful. Despite these being low resolution structures, as this reviewer had pointed out, the electron density around the interaction interface is good enough to confidently model the overall mode of intermolecular interactions. Mutants designed based on this structure, particularly CAL1 W22R mutation in isolation and in combination with F29R could perturb the interaction with CENP-A/H4 *in vitro* and in cells.

(Minor Quibbles)

R1-9. *Needs some more proofreading, in several spots amino acid #s contradict text/text or text/figure, a couple are just typos in the truncation numbers but at least 1 crucial interaction interface residue is numbered wrong when referring to figure 6A.*

Thank you for bringing this to our attention. This has been corrected.

R1-10. *Overall the discussion of the structure with histones is a bit long.*

We thank the reviewer for this suggestion. This section is now more concise.

R1-11. *There are NO page numbers.*

Page numbers have now been added.

R1-12. *Molecular weights from SEC-MALS and AUC are reported without any confidence statistics*

The mean standard error for the SEC-MALS data was reported in the materials and methods section. However, in hindsight we realised that it was not obvious to the reader so have now added them to the figures and stated them in the text. The 95% confidence intervals for AUC have also been added to the text on page 16.

R1-13. *Summary: this statement is inaccurate 'using crystal structures, we show how Drosophila CAL1, an evolutionarily distinct CENP-A chaperone, targets CENP-A to the centromere receptor CENP-C without the requirement of the Mis18 complex'. The crystal structure shows nothing about targeting, that is the interpretation.*

This sentence has been changed and now reads “Here, using crystal structures, we show how *Drosophila* CAL1, an evolutionarily distinct CENP-A chaperone, binds both CENP-A and the centromere receptor CENP-C without the requirement of the Mis18 complex”

R1-14. *Maybe a bit over stated 'This observation is in striking agreement with the subunit stoichiometry of the human pre-nucleosomal CENP-A/H4 in complex with HJURP' - whats so striking about that?*

Given the difference in the composition of molecular players involved in the CENP-A loading pathway in flies compared to other organisms and the lack of detectable sequence similarity among functionally equivalent molecular players, we found the similarity in the subunit stoichiometry of CAL/CENP-A/H4 and HJURP/CENP-A/H4 striking. However, we have now changed this statement as follows: 'This observation agrees with the subunit stoichiometry of the human pre-nucleosomal CENP-A/H4 in complex with HJURP'.

R1-15. *This statement in the discussion is strange, in light of the similarities with HJURP: The overall mode of CENP-A/H4 recognition by CAL1 appears to be novel'; especially since they state in the text: 'Structural superposition of CAL1-CENP-A/H4 onto its respective human and Kluyveromyces lactis structures, HJURP-CENP-A/H4 (PDB: 3R45) (Hu et al., 2011) and Scm3-CENP-A/H4 (PDB: 2YFV) (Cho and Harrison, 2011) showed that CAL1 employs a broadly similar mode of CENP-A recognition with a few striking differences (Fig. 3A). This needs to be phrased more carefully.*

We agree that this could cause confusion. We have now changed the text to read “When compared to the available ‘histone variant’ – chaperone complex crystal structures, the overall mode of CENP-A/H4 recognition by CAL1 appears to be novel as it is the only one which wraps around CENP-A/H4 through multiple CENP-A and H4 contacts resulting in the shielding of CENP-A/H4 surfaces involved in CENP-A/H4 tetramerisation, DNA binding and H2A/H2B binding - all critical for nucleosome assembly.”

Referee #2

The centromere is specified by sequence-independent epigenetic mechanisms, and histone H3 variant CENP-A plays an epigenetic mark for centromere specification. Therefore, it is a central subject in the centromere research field how CENP-A is deposited into centromeres. In human, CENP-A deposition is mediated by the CENP-A-specific chaperone HJURP and the chromatin licensing factor Mis18 complex. However, in *Drosophila*, there are no HJURP and Mis18 complex, although CENP-A is conserved. Instead, *Drosophila* specific factor Cal1 seems to mediate CENP-A deposition in *Drosophila* cells. However, it remains largely unknown how Cal1 mediates CENP-A deposition in *Drosophila* cells. In this study, authors analyzed crystal structures of the Cal1-CENP-A-H4 complex and demonstrated that the Cal1 N-terminal region binds to mainly CENP-A $\alpha 2$ helix, as HJURP does. However, the binding mode of Cal1 to CENP-A is slightly different from that of HJURP. Important residues for Cal1-CENP-A-H4 interaction observed in in vitro studies were confirmed by in vivo analyses.

In addition to the Cal1-CENP-A-H4 complex, authors also solved the structure of the Cal1-CENP-C complex in which the Cal1 C-terminal region binds to CENP-C dimer domain (Cupin domain). Combining two fine structures (Cal1-CENP-A-H4 and Cal1-CENP-C), the authors propose that Cal1 plays a critical function for CENP-A deposition in *Drosophila* cells. Overall, the structural studies are solid. Although previous studies predicted that Cal1 has similar activity of HJURP, this study clearly demonstrate that Cal1 binds to CENP-A-H4 using a similar way as HJURP at atomic resolution. It is an important step in the centromere research field. The structure of Cal1-CENP-C structure also gives us important information. We highly appreciate the challenge to the authors. However, I have several

technical concerns, and the authors should address them prior to publication. Especially, I do not think that experiments with expression of *Drosophila* proteins in U2OS cells are *in vivo* studies. If they want to examine protein interactions *in vivo*, they should use *Drosophila* cells.

We thank the reviewer for the positive evaluation of our work and for the constructive suggestions.

R2-1. As mentioned above, the experiments in Fig2D and Fig7D are not real *in vivo* studies. As other centromere (and/or regulatory) factors might not interact with *Drosophila* CENP-A, Cal1, and CENP-C in U2OS cells, these experiments are essentially similar to *in vitro* studies. Experiments themselves are slightly messy. If the authors want to demonstrate interaction *in vivo*, they should use *Drosophila* culture cells. Alternatively, they can show important residues in flies.

We thank the reviewer for this suggestion. We chose U2OS cells with a LacO array to validate our structure-guided mutants as it exploits the evolutionary divergence of human and *Drosophila* centromere components, resulting in a system with a minimal cellular interference as compared to the *Drosophila* cells. Our previous evaluation of the *Drosophila* centromere components with this system has recently proven powerful in elucidating the molecular sufficiency of CAL1, CENP-A and CENP-C in establishing a self-propagating epigenetic loop (Roure et al., 2019). However, as this reviewer had suggested, we have now extended the validation of structure-guided mutants in *Drosophila* culture cells (S2) with LacO arrays integrated in a chromosome arm. Consistent with our tethering experiments carried out in the U2OS cells, association of CAL1 mutants, designed to perturb CENP-A/H4 interaction, F43R and W22/F29A were significantly reduced at the tethering site. Likewise, the CENP-C mutants, L1357/M1407E and F1324R, designed to perturb the dimerization of cupin domain and CAL1 binding, respectively, both failed to interact with CAL1 at the tethering site. This data has now been added to revised Fig 3B & Fig 8E. We would also like to note that we did explore the possibility of evaluating the structure-guided mutants in siRNA/rescue in S2 cell. However, CAL1, CENP-A and CENP-C heavily depend on each other for their stability and function in *Drosophila* cells. To achieve specific depletion of CAL1 or CENP-C, double siRNA needs to be performed in combination with depleting PPA, a F-BOX protein involved in ubiquitin mediated degradation of CENP-A in flies (Moreno-Moreno et al., 2011), and CAL1 or CENP-C (Chen et al., JCB 2014). Hence, we thought a clean siRNA/rescue would be very difficult to achieve and results hard to interpret and decided not to proceed with this. In light of this, we have also limited the use of the term '*in vivo*' and instead used the phrase 'in cells'.

Figure 3B

Figure 8E

R2-2. Concerning section "Cal1 uses conserved and adaptive interaction...", authors mentioned "CENP-A at position 86". Is this true? It might be 186. In addition, there are no alignments of human, yeast CENP-A and Dm CENP-A. Author should show these alignments, if they discuss about conservation of human yeast and Drosophila CENP-A.

We thank the reviewer for pointing this out and making this suggestion. This typo has now been corrected and a new sequence alignment of human, yeast and *dm* CENP-A have been performed. CAL1 M186 is replaced by an Ala at the respective position in both human and yeast CENP-A. The multiple sequence alignment is now included in the revised Fig 4B.

Figure 4B

R2-3. Authors demonstrate that the $\alpha 1$ helix of Cal1 is swinging away compared with HJURP and $\alpha 5$ and $\alpha 6$ help binding of Cal1 to CENP-A-H4. This is a really interesting observation. Can they compare affinity of Cal1 to CENP-A-H4 in the presence or absence of $\alpha 5$ and $\alpha 6$? I think that Cal1 (1-50) might lack $\alpha 5$ and $\alpha 6$. How about its affinity compared with Cal1 (1-160)?

We thank the reviewer for this suggestion. This has proven technically difficult for the following reasons: making suitable quantities of homogeneous, well behaved folded *Drosophila* CENP-A/H4 is hard to achieve, and most importantly, reconstituted CENP-A/H4 requires high ionic strength buffer (at least 1M NaCl) for solubility. It also has a tendency to precipitate during concentration and is thus difficult to get to high concentrations. This coupled to the high ionic strength required for the reconstitution of the CENP-A/H4 and the resultant large enthalpies of dilution and ionisation meant that we could not realistically employ calorimetric binding methods such as ITC to probe this interaction. We tried to assess the difference in CENP-A/H4 binding affinity for His-CAL1₁₋₅₀ and His-CAL1₁₋₁₆₀ in Surface Plasmon Resonance (SPR) experiments. Although we found we could immobilise low levels of His-CAL1₁₋₅₀ or His-CAL1₁₋₁₆₀ constructs on Ni²⁺-primed NTA surfaces, it proved to be difficult and

challenging to get reproducible and stable levels, especially with the ionic conditions required to maintain the coherency of CENP-A/H4. Although the amount of immobilized His-CAL1₁₋₅₀ or His-CAL1₁₋₁₆₀ on the surface was sub-optimal, the experiments performed were good enough to suggest that both His-CAL1₁₋₅₀ or His-CAL1₁₋₁₆₀ bind CENP-A/H4 with a similar affinity under the “best” conditions we were able to test (shown in Fig R2, prepared for the purpose of this rebuttal). The normalised binding response (to account for the differences in the molecular weight of the two fragments), shows very little difference between the association and dissociation phases for either His-CAL1₁₋₅₀ or His-CAL1₁₋₁₆₀ interacting with CENP-A/H4. However, a very significant amount of optimisation of the run conditions and surface generation are required before definitive values could be assigned to the affinity and rate constants for this interaction.

We also attempted to probe if His-CAL1₁₋₅₀ and His-CAL1₁₋₁₆₀ bind CENP-A/H4 with varying affinity by assessing the salt sensitivity of the reconstituted complexes (His-CAL1₁₋₅₀/CENP-A/H4; His-CAL1₁₋₁₆₀/CENP-A/H4). We immobilised pre-formed complexes onto the Ni-NTA beads, washed the bound complex with buffers containing either 300mM NaCl or 3M NaCl in separate experiments and assessed the amount of CENP-A/H4 bound to CAL1 fragments by estimating the band intensities of Coomassie stained gels. We noticed that high salt wash displaced some immobilised CAL1 and associated CENP-A/H4 from Ni-NTA beads, but did not noticeably affect the relative amounts of CENP-A/H4 bound to different CAL1 fragments (Fig R3).

Although our efforts do not show noticeable difference in CENP-A/H4 binding affinity between CAL1₁₋₅₀ and His-CAL1₁₋₁₆₀, we cannot rule out that in a cellular context post-translational regulation such as phosphorylation or/and other intermolecular interaction involving the downstream helical segments of CAL1 might modulate CENP-A/H4 binding dynamics. We believe this is a plausible scenario as CAL1₁₋₅₀ is not sufficient to recruit CENP-A in cells (Chen et al., 2014 and Roure et al., 2019). Considering the preliminary nature of this data and space constraints we have decided not to include this data, but, have included a discussion on Page 10.

R2-4. *Crystal structures of S. pombe and Drosophila Cupin domain were reported recently (Chik JK et al., JBC, 2019, PMID: 31366733). Authors should cite this reference and discuss similarities and*

differences. In addition, the *S. pombe* Cupin domain binds to Moa1, and authors should mention whether similar binding region is used for Cal1 binding for the CENP-C Cupin domain.

We thank the reviewer for this suggestion. We have now compared our CENP-C cupin domain structure with the one determined by Chik JK et al., 2019. Both structures are nearly identical and superpose well with an RMSD of 0.27 Å. We have also mapped the *S. pombe* CENP-C cupin residues identified to be critical for Moa1 binding onto our *dm* CENP-C cupin domain bound to CAL1. The Moa1 binding site on *pombe* CENP-C cupin domain identified by Chik JK et al, appears extended with critical residues laterally spread across the deep pocket that forms the core of the cupin domain. Whereas, CAL1 binds at the periphery of the equivalent pocket and extends away from the pocket and towards the dimeric interface. Overall, this structural comparison suggests that *pombe* and *dm* cupin domains employ different modes of recognition to bind Moa1 and CAL1, respectively (revised Fig EV5B & EV5D). We have now referred to this work on page 14 and discussed the structural differences/similarities on page 19/20.

Figure EV5B

Figure EV5D

R2-5. While I understand expression of full length Cal1 is really hard, I am very curious whether Cal1 binds to CENP-A and CENP-C simultaneously. Although it might be hard to test this with in vitro experiments, if they can express tagged Cal1 in *Drosophila* cells and perform IP experiments, they might address this point. If they added such a experiments, quality of the paper would increase.

We thank the reviewer for this suggestion. We agree that it would be useful to show that CAL1 can bind CENP-A and CENP-C simultaneously. However, we feel that IP experiments would not be able to distinguish between the pools of CAL1 bound to both CENP-A and CENP-C and CAL1 bound to either CENP-A or CENP-C. Unfortunately, our efforts to generate recombinant full length CAL1 in bacteria or insect cells were not successful as CAL1 is a 979 aa long protein that is predicted to be predominantly unstructured. To overcome this, we generated several bonsai versions of CAL1 fusing the N-terminal CENP-A/H4 binding region and the C-terminal CENP-C binding region with a GSSGGSSG linker. We expressed and purified CAL1_{1-160-LL-841-979} and successfully reconstituted a hetero oligomeric complex containing CAL1_{1-160-LL-841-979}, CENP-A, H4 and CENP-C as analysed by the Size Exclusion Chromatography. The SEC data and corresponding SDS-PAGE has now been added in Fig 9B.

B

Figure 9B

Referee #3

This manuscript provides novel structural insights into the molecular interactions between the three key chromatin components of the *Drosophila* centromere; CENP-A, its chaperone CAL1 and the CENP-A and CAL1 binding protein CENP-C. These three proteins have been proposed to engage in self-enforcing tripartite positive feedback loop that may be sufficient, or is central to, centromere maintenance in fly cells. Some of the key interactions have been mapped earlier and proposed in work by the same team in a recent report that is published as a preprint. What they now add is further crystallographic characterization of the CAL1 domain that binds to CENP-A/H4 dimers. While the crystals obtained offered a modest resolution, they allowed to trace the backbone of CAL1 across most of the interaction surface on the CENP-A dimer. This provides insight into the conserved nature of CENP-A binding to its chaperone. While structures have been solved for both human and yeast proteins, this addition is relevant as CAL1 is evolutionally diverged, yet as is shown here, interacts with CENP-A prenucleosomes in a quite similar manner. The authors further show sedimentation data to determine protein stoichiometries and show a single CAL1 protein binds a single CENP-A/H4 heterodimer. Mutational analysis identified several key residues, although it is clear from this that the interaction of CAL1 with the histone dimer is multivalent, and many sites need to be altered to impact binding. Further, the structure of the CENP-C cupin domain was solved that was previously shown to interact with CAL1. Of interest here is that the binding of CAL1 requires the CENP-C cupin domain to dimerize and mutations in the dimerization domain of CENP-C abrogate CAL1 binding, both in vitro and on LacO arrays in cells. The work here offers the structural basis for this. While I am not a crystallographer, and cannot comment on the quality of the molecular models derived from the densities that were obtained, I believe that the work is both novel and very insightful offering new insight into the mechanisms of how CENP-A chromatin can be self-templating and epigenetically maintained.

The work is a comprehensive body of work combining X-ray crystallography, in vitro binding and stoichiometry analysis, in vivo binding assays and mutational analysis that make a convincing case for their claims.

I support publication overall but have several comments that need to be addressed.

We thank the reviewer for the positive evaluation of our work and for the constructive suggestions.

R3-1. Figure 2A, F43 is discussed in the text but not indicated in the figure. It would be helpful to have this shown.

We thank the reviewer for this suggestion. We agree that it would be useful for the reader to be able to easily identify the location of residue F43 of CAL1 on the structure. This has now been added for clarity to Fig 2A.

Figure 2A

R3-2. *Figure 2C: Is there a particular reason why the W22R and F29A/R single mutants are not tested?*

For completeness, as suggested by this reviewer, we have now generated a series of new CAL1 point mutants, W22A, W22R, F29A and F29R and expressed these as recombinant proteins. We then evaluated the effects of these CAL1 mutants on CENP-A/H4 binding in Ni-NTA assays (Fig 2C). These assays have been performed in replicates and effect on binding was estimated by calculating the ratio of the band intensities of CENP-A/H4 and CAL1 (revised Fig EV4D). Binding differences were assessed for statistical significance using the Mann-Whitney test. This revealed that F43R had a modest reduction in CENP-A/H4 binding, whilst W22R, W22A/F29A and W22R/F29R had a greater effect on CAL1's ability to interact with CENP-A/H4.

Figure 2C

Figure EV4D

R3-3. *Figure 3. The CAL1 CENP-/H4 structure is compared with corresponding structures of HJURP and Scm3. It would help to indicate what fragments of these proteins were resolved in these structures. Is it possible that some of the difference observed with CAL1 are due to the fact that different sized fragments are compared? This should be acknowledged.*

We agree that this is useful information for the reader to know. We have now carefully analysed the crystal structure of HJURP–CENP-A/H4 (PDB: 3R45; Hu et al., 2011) and Scm3–CENP-A/H4 (PDB: 2YFV; Cho & Harrison, 2011). We note that HJURP and Scm3 fragments used for structure analysis are shorter than the CAL1 fragment used here. We also performed secondary structure prediction

analysis of HJURP and Scm3 which suggest a lack of a similar α -helical segments downstream of $\alpha 1$ that stacks against CENP-A $\alpha 2$. So, we conclude that the contribution of downstream helical segments CAL1 for CENP-A/H4 binding is unique to flies. We now discuss this on page 19.

R3-4. *Figure 3C. The amount of CENP-A/H4 in the input is quite variable which makes interpretation of this experiment difficult. A repeat of this with more normalized starting material should be performed and quantification of the binding should be included.*

We thank the reviewer for these suggestions. These assays have now been performed in replicates with more normalised inputs (Fig 4D). Effect of CENP-A mutations on CAL1 binding were estimated by calculating the ratio of the band intensities between CENP-A/H4 and CAL1 (revised figure Fig EV4D). Significance of binding differences was analysed using an unpaired two tailed *t*-test. This revealed that S154Q/M186A double mutation in CENP-A/H4 had a modest reduction in binding CAL1₁₋₁₆₀, whilst S154Q/M186A/Q190G triple mutation had a greater effect on CAL1's ability to interact with CENP-A/H4.

R3-5. *The models in figure 4 are very busy and it is hard to see the steric clashes that CAL1 would cause. I suggest including here images of just the CENP-A/H4 heterotetramer (no H2A and H2B and no DNA) to visualize better how CAL1 prevents CENP-A/H4 tetramerization.*

We thank the reviewer for this feedback. A new panel has now been added to Fig 5 in line with the reviewer's suggestions (shown below).

Figure 5

R3-6. *Figure 5 would be more legible if a linear diagram of CENP-C is shown mapping the domain that is solved and showing the position of the Cupin domain (similar to the schematics in Figure 1A).*

We thank the reviewer for this suggestion. This has now been added to Fig 6A.

Figure 6A

R3-7. *Figure 7. The CENP-C residues M1407 and L1357 are important for CENP-C dimerization and in this figure they are shown to be also critical for CAL1 binding. The authors interpret this as dimerization of CENP-C being required for CAL1 binding. Alternatively, the CENP-C mutations directly affect CAL1 interactions. How are these residues positioned in the structure relative to the CAL1 binding site? Is it possible that the M1407E and L1357E directly affect CAL1 interactions? If not, this can be highlighted in the figure.*

We thank the reviewer for highlighting this important point. The residues M1407 and L1357 are buried within the dimerization domain and would not be available on the surface to interact with CAL1 (shown in Fig 6D). Therefore, mutating them would only disrupt the dimerization, not the interaction site which is situated near the dimerization interface (shown in Fig 6A). We have now highlighted these residues in revised Fig 6D by encircling them.

Fig 6D

R3-8. *Halfway through the discussed it is stated that CENP-C is present in all organisms with monocentric chromosomes. However, kinetoplasts have monocentric, regional centromeres but lack CENP-C.*

This sentence has been modified to correct this. It now reads “However, CAL1 seems to compensate for this loss by directly associating with CENP-C, which is present in most organisms with monocentric chromosomes”.

R3-9. *While the single CAL1 binding to a CENP-C dimer is very interesting as it may provide specificity for the centromere, the idea that CAL1 can bring in CENP-C in turn is more vague. The*

authors suggest that CAL1 dimers or oligomers may bring in CENP-C. However in their previous work (Roure 2019, bioRxiv) they show that CAL1 can bring CENP-C to chromatin but only in the presence of CENP-A. This would suggest that the role for CAL1 in CENP-C recruitment is indirect via CENP-A nucleosome. It is therefore not clear whether the CAL1 CENP-C interaction is sufficient to recruit CENP-C to chromatin. While the authors may choose to discuss this as a hypothetical scenario, I'm very much opposed to the term CENP-C "loading" or CAL1 being a CENP-C "loader". Loading implies more than just binding. It indicates a multistep process to transition from one state to another. There is no evidence that CAL1 is a chaperone for CENP-C that needs to be handed off to another binding partner or DNA. CENP-C simply binds to CENP-A and it binds to CAL1. I urge to authors to stick to the word "binding" not loading.

We thank the reviewer for this feedback. We have now edited the text to remove the reference to 'loading'

Thank you for submitting your revised manuscript for our consideration. It has now been re-reviewed by all three original referees (see comments below), in light of whose positive assessment I am pleased to inform you that we have now accepted the study for publication in The EMBO Journal!

REFEREE REPORTS

Referee #1:

We are happy with the changes and the response to our critiques. Congratulations to the authors for a very nice manuscript

Referee #2:

I found that authors seriously addressed all concerns raised by all reviewers. The paper has been improved and has a quality. I now recommend this for publication.

Referee #3:

This is a revised version of a previously submitted paper. The authors have made a significant effort to accommodate my criticisms. They have added additional data, particularly on single point mutants in Cal1 affecting CENP-A/H4 binding as well as new binding assays with CENP-A mutants. They have adequately addressed virtually all my other concerns. I have no further comments and support publication.

Corresponding Author Name: A. Arockia Jeyaprakash

Journal Submitted to: Embo J

Manuscript Number: EMBOJ-2019-103234